# Differentially private partitioned variational inference

**Mikko A. Heikkilä**                                              *mikkoaaro.heikkila@telefonica.com*
*Telefónica Research*

**Matthew Ashman**                                                          *mca39@cam.ac.uk*
*Department of Engineering*
*University of Cambridge*

**Siddharth Swaroop**                                              *siddharth@seas.harvard.edu*
*School of Engineering and Applied Sciences*
*Harvard University*

**Richard E. Turner**                                                        *ret26@cam.ac.uk*
*Department of Engineering*
*University of Cambridge*

**Antti Honkela**                                                      *antti.honkela@helsinki.fi*
*Department of Computer Science*
*University of Helsinki*

**Reviewed on OpenReview:** *https://openreview.net/forum?id=55BcghgicI*

## Abstract

Learning a privacy-preserving model from sensitive data which are distributed across multiple devices is an increasingly important problem. The problem is often formulated in the federated learning context, with the aim of learning a single global model while keeping the data distributed. Moreover, Bayesian learning is a popular approach for modelling, since it naturally supports reliable uncertainty estimates. However, Bayesian learning is generally intractable even with centralised non-private data and so approximation techniques such as variational inference are a necessity. Variational inference has recently been extended to the non-private federated learning setting via the partitioned variational inference algorithm. For privacy protection, the current gold standard is called differential privacy. Differential privacy guarantees privacy in a strong, mathematically clearly defined sense.

In this paper, we present differentially private partitioned variational inference, the first general framework for learning a variational approximation to a Bayesian posterior distribution in the federated learning setting while minimising the number of communication rounds and providing differential privacy guarantees for data subjects.

We propose three alternative implementations in the general framework, one based on perturbing local optimisation runs done by individual parties, and two based on perturbing updates to the global model (one using a version of federated averaging, the second one adding virtual parties to the protocol), and compare their properties both theoretically and empirically. We show that perturbing the local optimisation works well with simple and complex models as long as each party has enough local data. However, the privacy is always guaranteed independently by each party. In contrast, perturbing the global updates works best with relatively simple models. Given access to suitable secure primitives, such as secure aggregation or secure shuffling, the performance can be improved by all parties guaranteeing privacy jointly.

# 1 Introduction

Communication-efficient distributed methods that protect user privacy are a basic requirement for many machine learning tasks, where the performance depends on having access to sensitive personal data. Federated learning (Brendan McMahan et al., 2016; Kairouz et al., 2019) is a common approach for increasing communication efficiency with distributed data by pushing computations to the parties holding the data, thereby leaving the data where they are. While it has been convincingly demonstrated that federated learning by itself does not guarantee any kind of privacy (Zhu et al., 2019), it can be combined with differential privacy (DP, Dwork et al. 2006b; Dwork & Roth 2014), which does provide formal privacy guarantees.

In settings which require uncertainty quantification as well as high prediction accuracy, Bayesian methods are a natural approach. However, Bayesian posterior distributions are often intractable and need to be approximated. Variational inference (VI, Jordan et al. 1999; Wainwright et al. 2008) is a well-known and widely used approximation method based on solving a related optimisation problem; the most common formulation minimises the Kullback-Leibler divergence between the approximation and the true posterior.

In this paper, we focus on privacy-preserving federated VI in the cross-silo setting (Kairouz et al., 2019). We consider a common and important setup, where the parties (or 'clients') have sensitive horizontally partitioned data, i.e., local data with shared features, and the aim is to learn a single model on all the data. Such a setting arises, for example, when several hospitals want to train a joint model on common features without sharing their actual patient data with any other party. Our main problem is to learn a posterior approximation from the partitioned data under DP, while trying to minimise the amount of server-client communications.

We propose a general framework to solve the problem based on partitioned variational inference (PVI, Ashman et al. 2022). On a conceptual level, the main steps of PVI are the following: i) server sends current global model to clients, ii) clients perform local optimisation using their own data, iii) clients send updates to server, iv) server updates the global approximation. In our solution, called differentially private partitioned variational inference (DP-PVI), the clients enforce DP either independently or, given access to some suitable secure primitive, jointly with the other clients. We consider three different implementations of DP-PVI, one based on perturbing the local optimisation at step ii) of PVI (called DP optimisation), and two on perturbing the model updates at step iii) of PVI (called local averaging and virtual PVI clients).

Crucially, we empirically demonstrate that with our approaches the number of communication rounds between the server and the clients can be kept significantly lower than in the existing DP VI baseline solution, that requires communicating gradients for each regular VI optimisation step, while achieving nearly identical performance in terms of accuracy and held-out likelihood. Additionally, while the baseline requires communicating the gradients using some suitable secure primitive to achieve good utility, our approaches do not require any such primitives, although they can be easily combined with two of our approaches (local averaging and virtual PVI clients).[1] Finally, compared to the communication minimising baselines given by DP Bayesian committee machines, that require a single communication round between the server and the clients, our solutions provide clearly better prediction accuracies and held-out likelihoods in a variety of settings.

**Our contribution**   Our main contributions are the following:

- We introduce DP-PVI, a communication-efficient general approach for DP VI on horizontally partitioned data.

- Within the general framework, we propose three differing implementations of DP-PVI, one based on perturbing local optimisation, termed DP optimisation, and two based on perturbing model updates, that we call local averaging and virtual PVI clients.

- Compared to the baseline of standard (global) DP VI, we experimentally show that our proposed implementations need orders of magnitude fewer server-client communication rounds, and can be

---

[1]In this paper, instead of considering any specific secure primitive implementation, such as a secure aggregator or a secure shuffler, we assume access to a black-box trusted aggregator in the comparisons.

trained without any secure primitives, while achieving comparable model utility under various settings. Compared to the DP Bayesian committee machine baselines, all our implementations can significantly improve on the resulting approximation quality under various models and datasets.

- We compare the relative advantages and disadvantages of our methods, both theoretically and experimentally, and make recommendations on how to choose the best method based on the task at hand:

  - We demonstrate that no single implementation outperforms the others in all settings.
  - We show that DP optimisation works well when there is enough local data available, on both simple and complex models. However, it not benefit from access to secure primitives. It can therefore lag behind the other methods when there are many clients without much local data, but with access to a trusted aggregator.
  - In contrast, local averaging and virtual PVI clients work best with relatively simple models, but can struggle with more complex ones. Since they can directly benefit from access to a trusted aggregator, they can outperform DP optimisation in a setting with many clients, little local data on each, and a trusted aggregator available. We find that using virtual PVI clients tends to be more stable than local averaging.

## 2 Related work

In the non-Bayesian setting, federated learning, with and without privacy concerns, has seen a lot of recent research (Kairouz et al., 2019).

VI without privacy has been considered in a wide variety of configurations, a considerable proportion of which can be interpreted as specific implementations of PVI. With just a single client, the local free-energy is equivalent to the global free-energy and global VI is recovered (Hinton & Van Camp, 1993). PVI with multiple clients unifies many existing local VI methods, in which each factor involves a subset of datapoints and may also include only a subset of variables over which the posterior is defined. This includes variational message passing (Winn et al., 2005) and its extensions (Knowles & Minka, 2011; Archambeau & Ermis, 2015; Wand, 2014), and is related to conjugate-computation VI (Khan & Lin, 2017). When only a single pass is made through the clients, PVI recovers online VI (Ghahramani & Attias, 2000), streaming VI (Broderick et al., 2013) and variational continual learning (Nguyen et al., 2018). See Ashman et al. (2022) for a more detailed overview of the relationships between PVI and these methods.

There is a rich literature on Bayesian learning with DP guarantees in various settings. Perturbing sufficient statistics in exponential family models has been applied both with centralised data (Dwork & Smith, 2010; Foulds et al., 2016; Zhang et al., 2016; Honkela et al., 2018) as well as with distributed data (Heikkilä et al., 2017). In the centralised setting, Dimitrakakis et al. (2014) showed that under some conditions, drawing samples from the true posterior satisfies DP. Posterior sampling under DP has been extended (Zhang et al., 2016; Geumlek et al., 2017; Dimitrakakis et al., 2017), and generalised also based on, e.g., Langevin and Hamiltonian dynamics (Wang et al., 2015; Li et al., 2019; Räisä et al., 2021) as well as on general Markov chain Monte Carlo (Heikkilä et al., 2019; Yıldırım & Ermiş, 2019). As an orthogonal direction for combining Bayesian learning with DP, Bernstein & Sheldon (2018) proposed a method for taking the DP noise into account when estimating the posterior with exponential family models to avoid model overconfidence.

DP-VI has been previously considered in the centralised setting. Jälkö et al. (2017) first introduced DP-VI for non-conjugate models based on DP-SGD, while Foulds et al. (2020) proposed a related variational Bayesian expectation maximization approach based on sufficient statistics perturbation for conjugate exponential family models.

Also in the centralised setting, Vinaroz & Park (2021) recently proposed DP stochastic expectation propagation, an alternative approximation method to VI that also has some close ties to the PVI framework (see e.g. Ashman et al. 2022), based on natural parameter perturbation. While there are several technical differences in how DP is guaranteed, and Vinaroz & Park (2021) do not discuss the distributed setting or propose a federated algorithm, we would expect that there is no fundamental reason why their approach could not be made to

work in the federated setting as well, given enough changes to the centralised algorithm. With reasonable privacy parameters, we would expect comparisons to reflect the general properties of the underlying non-DP approaches (see e.g. Minka 2005; Li et al. 2015; Ashman et al. 2022 and the references therein for a discussion on the properties of non-DP variational and EP-style algorithms). The application of such methods to the private federated setting would be an interesting direction for future work.

In the non-Bayesian DP literature, the basic idea in our local averaging and virtual client approaches is close to the subsample and aggregate approach proposed by Nissim et al. (2007). In the same vein as our local averaging, Wei et al. (2020) proposed training a separate model for each single data point and combining the models by averaging the parameters in the context of learning neural network weights under DP. They do not consider other possible partitions or the trade-off between DP noise and the estimator variance.

## 3 Background

In this section we give a short overview of the most important background knowledge, starting with PVI in Section 3.1 and continuing with DP in Section 3.2.

### 3.1 Partitioned variational inference (PVI)

In Bayesian learning, we are interested in the posterior distribution

$$p(\theta|x) \propto p(x|\theta)p(\theta),$$

where $p(\theta)$ is a prior distribution, $p(x|\theta)$ is a likelihood, $x \in \mathcal{X}^n$ is some dataset of size $n$, and $\theta$ are the model parameters. Note that these are all different distributions, but we overload the notation in a standard way and identify the distributions by their arguments to keep the writing less cumbersome. For example, instead of $p_\theta(\theta)$ we simply write $p(\theta)$ for the prior. In this paper, we typically have $\theta \in \mathbb{R}^d$ for some $d$, and each element $x_i \in \mathbb{R}^{d'}, i = 1, \ldots, n$ for some $d'$.

When the posterior is in the exponential family of distributions, it is always tractable (see, e.g., Bernardo & Smith 1994):

**Definition 1.** *A distribution over $x \in \mathcal{X}^n$, indexed by a parameter vector $\theta \in \Theta \subset \mathbb{R}^d$ is an exponential family distribution, if it can be written as*

$$p(x|\theta) = h(x) \exp\left(T(x) \cdot \eta(\theta) - A(\eta(\theta))\right) \tag{3.1}$$

*for some functions $h : \mathcal{X}^n \to \mathbb{R}, T : \mathcal{X}^n \to \mathbb{R}^d, A : \Theta \to \mathbb{R}$. When $\eta(\theta) = \eta$, the parameters $\eta$ are called natural parameters, $T$ are sufficient statistics, $A$ is the log-partition function, and $h$ is a base measure.*

When the posterior is not in the exponential family, however, we need to resort to approximations. VI is a method for approximating the true posterior by solving an optimization problem over some tractable family of distributions (see e.g. Jordan et al. 1999 for an introduction to VI and Zhang et al. 2019 for a survey of more recent developments).

Writing $q(\theta|\lambda)$ for the approximating distribution parameterised with variational parameters $\lambda \in \mathbb{R}^{d_{VI}}$ for some $d_{VI}$, the main idea in VI is to find optimal parameters $\lambda_{VI}^*$ that minimise some notion of distance between the approximation and the true posterior, with the most common choice being Kullback-Leibler divergence:

$$\lambda_{VI}^* = \underset{q \in \mathcal{Q}}{\arg\min} \left[D_{\mathrm{KL}}(q(\theta|\lambda)\|p(\theta|x))\right], \tag{3.2}$$

where $\mathcal{Q}$ is some tractable family of distributions. However, since the optimisation problem in Equation 3.2 is usually still not easy-enough for solving directly, the actual optimisation is typically done by maximising the so-called evidence lower bound (ELBO, also called negative variational free-energy):

$$\lambda_{VI}^* = \underset{q \in \mathcal{Q}}{\arg\max} \left[\mathbb{E}_q[\log p(x|\theta)] - D_{\mathrm{KL}}(q(\theta|\lambda)\|p(\theta))\right]. \tag{3.3}$$

It can be shown that the optimal solution $\lambda_{VI}^*$ that solves Equation 3.3 also solves the original minimization problem in Equation 3.2.

In the setting we consider, there are $M$ clients with shared features, and client $j$ holds $n_j$ samples. In the PVI framework (Ashman et al., 2022), this federated learning problem is solved iteratively. We start by defining the following variational approximation:

$$q(\theta|\lambda) = \frac{1}{Z_q} p(\theta) \prod_{j=1}^{M} t_j(\theta|\lambda_j) \simeq \frac{1}{Z} p(\theta) \prod_{j=1}^{M} p(x_j|\theta) = p(\theta|x), \tag{3.4}$$

where $\lambda, \lambda_j$ are variational parameters, $Z_q, Z$ are normalizing constants, $x_j$ is the $j$th data shard, and $t_j$ are client-specific factors refined over the algorithm run. The basic structure of the general PVI algorithm is given in Algorithm 1.

Note that in Algorithm 1, during the local optimisation (Equation 3.5) the cavity distribution (Equation 3.6) works as an effective prior: the variational parameters for the factors $t_j, j \neq m$ are kept fixed to their previous values.

---

**Algorithm 1** Non-private PVI (Ashman et al., 2022)

---

**Require:** Number of global updates $S$, prior $p(\theta)$, initial client-specific factors $t_j^{(0)}, j = 1, \ldots, M$.
1: **for** $s = 1$ to $S$ **do**
2:     Server chooses a subset $b^{(s)} \subseteq \{1, \ldots, M\}$ of clients according to an update schedule and sends the current global model parameters $\lambda^{(s-1)}$.
3:     Each chosen client $m \in b^{(s)}$ finds a new set of parameters by optimising the local ELBO:

$$\lambda^* = \arg\max_{q \in \mathcal{Q}} \left[ \mathbb{E}_q[\log p(x_m|\theta)] - D_{\mathrm{KL}}(q(\theta|\lambda) \| p_{\backslash m}^{(s-1)}(\theta)) \right], \tag{3.5}$$

where $p_{\backslash m}^{(s-1)}$ is the so-called cavity distribution:

$$p_{\backslash m}^{(s-1)}(\theta) \propto p(\theta) \prod_{j \neq m}^{M} t_j(\theta|\lambda_j^{(s-1)}). \tag{3.6}$$

4:     Each chosen client sends an update $\Delta t_m^{(s)}(\theta)$, given by

$$\Delta t_m^{(s)}(\theta) \propto \frac{t_m(\theta|\lambda_m^*)}{t_m(\theta|\lambda_m^{(s-1)})} \propto \frac{q(\theta|\lambda^*)}{q(\theta|\lambda^{(s-1)})}, \tag{3.7}$$

to the server.
5:     Server updates the global model by incorporating the updated local factors:

$$q(\theta|\lambda^{(s)}) \propto q(\theta|\lambda^{(s-1)}) \prod_{m \in b^{(s)}} \Delta t_m^{(s)}(\theta).$$

6: **end for**
7: **return** Final variational approximation $q(\theta|\lambda^{(S)})$.

---

As a high-level overview, the PVI learning loop consists of the server sending current model to clients, the clients finding new local parameters via local optimisation, the clients sending an update to the server, and the server updating the global approximation.

Depending on the update schedule different variants of PVI are possible. In this paper, we use *sequential* PVI, where each client is visited in turn, and *synchronous* PVI, where all clients update in parallel (see Ashman et al. 2022 for more discussion on the PVI variants). The main idea in PVI is that the information from other

sites is transmitted to client $m$ via the other $t$-factors, while client $m$ runs optimisation with purely local data. This reduces the number of server-client communications by pushing more computation to the clients.

Ashman et al. (2022) show that PVI has several desirable properties that connect it to the standard non-distributed (global) VI. Most importantly, optimising the local ELBO as in Equation 3.5 can be shown to be equivalent to a variational KL optimisation, and a fixed point of PVI is guaranteed to be a fixed point of the global VI.

### 3.2 Differential privacy (DP)

DP is essentially a robustness guarantee for stochastic algorithms (see e.g. Dwork & Roth 2014 for an introduction to DP and discussion on the definition of privacy). Formally we have the following:

**Definition 2** (Dwork et al. 2006b;a). *Let $\varepsilon > 0$ and $\delta \in [0, 1]$. A randomised algorithm $\mathcal{A} : \mathcal{X}^n \to \mathcal{O}$ is $(\varepsilon, \delta)$-DP if for every neighbouring $x, x' \in \mathcal{X}^n$ and every measurable set $E \subset \mathcal{O}$,*

$$\Pr(\mathcal{A}(x) \in E) \leq \mathrm{e}^\varepsilon \Pr(\mathcal{A}(x') \in E) + \delta.$$

The basic idea in the definition is that any single individual should only have a limited effect on the output. When this is guaranteed, the privacy of any given individual is protected, since the result would have been nearly the same even if that individual's data had been replaced by an arbitrary sample. Definition 2 formalises this idea by requiring that the probability of seeing any given output is nearly the same with any closely-related input dataset (the neighbouring datasets $x, x'$). The actual level of protection depends on the privacy parameters $\varepsilon, \delta$: larger values mean less privacy.

The type and granularity of the privacy guarantee can be tuned by choosing an appropriate neighbourhood definition. Typical examples include sample-level ($x, x'$ differ by a single sample) and user-level ($x, x'$ differ by a single user's data) neighbourhoods. In this work, we use the bounded neighbourhood definition, which is also known as substitution neighbourhood, and assume that each individual has a single sample in the full combined training data, i.e., datasets $x, x'$ are neighbours, if $|x| = |x'|$, and they differ by a single sample. With these definitions, individual privacy guarantees correspond to sample-level DP.

DP has several nice properties as a privacy guarantee, but the most important ones for our purposes are composability (repeated use of the same sensitive data erodes the privacy guarantees in a controllable manner), and immunity to post-processing (if the output of a stochastic algorithm is DP, then any stochastic or deterministic post-processing results in the same or stronger DP guarantees).

We use the well-known Gaussian mechanism, that is, adding i.i.d. Gaussian noise with equal variance to each component of a query vector, as a basic privacy mechanism:

**Definition 3** (Gaussian mechanism, Dwork et al. 2006a). *Let $f : \mathcal{X}^n \to R^d$ be a function s.t. for neighbouring $x, x' \in \mathcal{X}^n$, there exists a constant $C > 0$ satisfying*

$$\sup_{x,x'} \|f(x) - f(x')\|_2 \leq C.$$

*A randomised algorithm $\mathcal{G} : \mathcal{G}(x) = f(x) + \xi$, where $\xi \sim \mathcal{N}(0, \sigma I_d)$ is called the Gaussian mechanism.*

When a privacy mechanism, e.g., the Gaussian mechanism, is run by first subsampling a minibatch of the full data, and running the mechanism using only the minibatch instead of the full data, the mechanism is referred to as a *subsampled mechanism*. For subsampling, we use sampling without replacement:

**Definition 4** (Sampling without replacement). *A randomised function $WOR_b : \mathcal{X}^n \to \mathcal{X}^b$ is a sampling without replacement subsampling function, if it maps a given dataset into a uniformly random subset of size $b$ of the input data.*

The main benefit for privacy when using data subsampling is the effect of privacy amplification, i.e., the additional randomisation due to the subsampling enhances the privacy guarantees depending on the subsampling method and the *subsampling fraction* given by $q_{sample} = \frac{b}{n}$, where $b$ is the minibatch size and $n$

is the total data size in Definition 4. Given a base mechanism $\mathcal{A}_\sigma$ and a minibatch size $b$, the subsampled mechanism using sampling without replacement is the combined mechanism $\mathcal{A}_\sigma \circ WOR_b$.

To quantify the total privacy resulting from (iteratively) running (subsampled) DP algorithms, we use the following privacy accounting oracle:

**Definition 5** (Accounting Oracle)**.** *An* accounting oracle *is a function $\mathbb{O}$ that evaluates $(\epsilon, \delta)$-DP privacy bounds for compositions of (subsampled) mechanisms. Specifically, given $\delta$, a sub-sampling ratio $q_{sample} \in (0, 1]$, the number of iterations $T \geq 1$ and a base mechanism $\mathcal{A}_\sigma$, the oracle gives an $\epsilon$, such that a $T$-fold composition of $\mathcal{A}_\sigma$ using sub-sampling with ratio $q_{sample}$ is $(\epsilon, \delta)$-DP, i.e.,*

$$\mathbb{O} : (\delta, q_{sample}, T, \mathcal{A}_\sigma) \mapsto \epsilon.$$

In the experiments, we use the Fourier accountant (Koskela et al., 2020) as an accounting oracle to keep track of the privacy parameters, since it can numerically establish an upper bound for the total privacy loss with a given precision level.

## 4 Differentially private partitioned variational inference

In the setting we consider, there are $M$ parties or clients connected to a central server, with client $j$ holding some amount $n_j$ of data (we assume there is exactly one sample per individual protected by DP in the full joint data) with common features (horizontal data partitioning). The clients do not want to share their data directly but agree to train a model given DP guarantees. The DP guarantees are enforced on a sample-level, that is, we assume that any given individual we want to protect has a single data sample that is held by exactly one client. The central server aims to learn a single model from the clients' data, while minimising the number of communication rounds between the server and the clients.

As discussed in Section 3.1, the PVI framework allows for effectively reducing the number of global communication rounds by pushing more computation to the clients. This also enables several options for guaranteeing DP on the client side, either by each client alone or jointly with the other clients via secure primitives.

We consider two general approaches the clients can use for enforcing DP in PVI learning:

1. Perturbing the local optimisation (step 3 in Algorithm 1),

2. Perturbing the model parameter updates (step 4 in Algorithm 1).

The first option, which we term *DP optimisation*, relies on the fact that at each optimisation step in Algorithm 1, the local ELBO in Equation 3.5 only depends on the local data at the given client. To guarantee DP independently of others, each client can therefore perturb the local optimisation with a suitable DP mechanism. In practice, this approach can be implemented, e.g., using DP-stochastic gradient descent (DP-SGD) as we show in Section 4.1.

For the second option, since a given client only affects the global model through the parameter updates at step 4 in Algorithm 1, each client can enforce DP by perturbing the update, either independently or jointly with the other clients. Besides the naive *parameter perturbation*, we propose two improved alternatives in Section 4.2. We call these approaches *local averaging* and adding *virtual PVI clients*.

In this paper, instead of considering any particular secure primitive like secure aggregation (see e.g. Shamir 1979; Rastogi & Nath 2010) or secure shuffling (see e.g. Chaum 1981; Cheu et al. 2019), we assume a black-box trusted aggregator capable of summing reals, where necessary. In these cases we also assume that the clients themselves are honest, i.e., they follow the protocol and do not try to gain additional information during the protocol run (or honest but curious, that is, they follow the protocol but will try to gain information such as actual noise values used for DP randomisation during the protocol run, with minor modifications to the relevant bounds). Any actual implementation would need to handle problems arising, for example, from finite precision (see e.g. Agarwal et al. 2021; Chen et al. 2022 and references therein for a discussion on implementing distributed DP). These considerations apply equally to all variants, and hence do not affect their comparisons or our main conclusions. We leave these issues to future work.

Table 1 highlights the most important DP noise properties of our proposed solutions: whether the DP noise level can be affected by the local data size (intuitively, we could hope that guaranteeing DP with plenty of local data gives better utility), and whether the approach can benefit from access to a trusted aggregator (this enables the clients to guarantee DP jointly, so the total noise level can be less than when every client enforces DP independently).

| | noise scale affected by local data size | benefit from a trusted aggregator |
|---|---|---|
| DP optimisation | ✓ | x |
| parameter perturbation | x | ✓ |
| local averaging | ✓ | ✓ |
| virtual PVI clients | ✓ | ✓ |

Table 1: Properties of DP-PVI approaches

In the rest of this section we state the formal DP guarantees for each approach and discuss their properties. For ease of reading, since the proofs are fairly straight-forward, all proofs as well as the properties of non-DP local averaging can be found in Appendix A.

## 4.1 Privacy via local optimisation: DP optimisation

To guarantee DP during local optimisation, one option is to use differentially private stochastic gradient descent (DP-SGD) (Song et al., 2013; Bassily et al., 2014; Abadi et al., 2016): for every local optimisation step, we clip each per-example gradient and then add Gaussian noise with covariance $\sigma^2 I$ to the sum. The formal privacy guarantees are stated in Theorem 6.

**Theorem 6.** *Running DP-SGD for client-level optimisation in Algorithm 1, using subsampling fraction $q_{sample} \in (0,1]$ on the local data level for $T$ local optimisation steps in total, with $S$ global updates interleaved with the local steps, the resulting model is $(\varepsilon, \delta)$-DP, with $\delta \in (0,1)$ s.t. $\varepsilon = \mathbb{O}(\delta, q_{sample}, T, \mathcal{G}_\sigma)$.*

*Proof.* See proof A.1 in the Appendix. □

Although DP-SGD in general is not guaranteed to converge, there are some known utility guarantees in the empirical risk minimization (ERM) framework, e.g., for convex and strongly convex loss functions (Bassily et al., 2014). It has also been empirically shown to work well on a number of problems with non-convex losses, such as learning neural network weights (Abadi et al., 2016).

In our setting, DP-SGD can directly benefit from increasing local data size on a given client via the sub-sampling amplification property: adding more local data while keeping the batch size fixed results in a smaller sampling fraction $q_{sample}$ and hence gives better privacy.

In contrast, when using DP-SGD with a limited communication budget, it is non-trivial to derive direct privacy benefits from adding more clients to the setting. This is the case even when we assume access to a trusted aggregator, since the gradients of the local ELBO in Equation 3.5 only depend on a single client's data.

## 4.2 Privacy via model updates

To guarantee DP when communicating an update from client $m$ to the server at global update $s$, we can clip and perturb the change in model parameters corresponding to $\Delta t_m^{(s)}$ at step 4 in Algorithm 1 directly. This naive *parameter perturbation* approach often results in having to add unpractical amounts of noise to each query, which severely degrades the model utility. The problem arises because the local data size in this case will typically have no direct effect either on the DP noise level or on the query sensitivity.

To improve the results by allowing the local data size to have a direct effect on the noise addition, we propose two possible approaches that generalise the naive parameter perturbation: i) *local averaging* and ii) adding *virtual PVI clients*. Both are based on partitioning the local data into non-overlapping shards and optimising a separate local model on each, but they differ on the objective functions and on how the local results are combined after training for a global model update. Additionally, virtual PVI clients with DP requires all virtual factors to be in a common exponential family.[2] As a limiting case, when using a single local data partition both methods are equivalent to the naive parameter perturbation.

Assuming a trusted aggregator, with both of our proposed methods we can scale the noise level with the total number of clients in the protocol using $\mathcal{O}(MS)$ server-client communications, where $M$ is the number of clients and $S$ the total number of global updates, the same number as running non-DP PVI with synchronous updates.

Next, we present the methods and show that they guarantee DP, starting with local averaging in Section 4.2.1 and continuing with virtual PVI clients in Section 4.2.2.

### 4.2.1 Local averaging

Algorithm 2 describes the main steps needed for running (non-private) PVI with local averaging.

---

**Algorithm 2** PVI with local averaging

---

1: Each client $m = 1, \ldots, M$ partitions its local data into $N_m$ non-overlapping shards.
2: **for** $s = 1$ to $S$ **do**
3:     Server chooses a subset $b^{(s)} \subseteq \{1, \ldots, M\}$ of clients according to an update schedule and sends the current global model parameters $\lambda^{(s-1)}$.
4:     Each chosen client $m \in b^{(s)}$ finds $N_m$ sets of new parameters by optimising the local objectives all starting from a common initial value (the previous global model parameters $\lambda^{(s-1)}$):

$$\lambda^*_{m_k} = \arg\max_{q \in \mathcal{Q}} \left[ \mathbb{E}_q[\log p(x_{m,k}|\theta)] - \frac{1}{N_m} D_{\mathrm{KL}}(q(\theta|\lambda) \| p_{\backslash m}^{(s-1)}(\theta)) \right], \quad k = 1, \ldots, N_m, \tag{4.1}$$

where $p_{\backslash m}^{(s-1)}$ is the cavity distribution as in Equation 3.6. The new parameters used for calculating an update for client $m$ in PVI are given by the local average:

$$\lambda^* = \frac{1}{N_m} \sum_{k=1}^{N_m} \lambda^*_{m_k}. \tag{4.2}$$

5:     Each chosen client sends the update $\Delta t_m^{(s)}(\theta)$, defined as

$$\Delta t_m^{(s)}(\theta) \propto \frac{t_m(\theta|\lambda^*_m)}{t_m(\theta|\lambda_m^{(s-1)})} \propto \frac{q(\theta|\lambda^*)}{q(\theta|\lambda^{(s-1)})},$$

    to the server.
6:     Server updates the global model by incorporating the updated local factors:

$$q(\theta|\lambda^{(s)}) \propto q(\theta|\lambda^{(s-1)}) \prod_{m \in b^{(s)}} \Delta t_m^{(s)}(\theta).$$

7: **end for**
8: **return** Final variational approximation $q(\theta|\lambda^{(S)})$.

---

[2]Non-DP PVI with synchronous updates has a similar restriction: all factors need to be in the same exponential family as the prior due to issues with proper normalization, see Ashman et al. 2022. Therefore, when using synchronous PVI server all our approaches, including DP optimisation, inherit this assumption as well.

Note that the objective in Equation 4.1 is the regular PVI local ELBO where the KL-term is re-weighted to reflect the local partitioning. This is equivalent to using the PVI objective with a tempered (cold) likelihood $p(x_{m,k}|\theta)^{N_m}$. In Appendix A, we show that PVI with local averaging has the same fundamental properties, with minor modifications, as regular PVI. For example, with local averaging the local ELBO optimisation is equivalent to a variational KL optimisation, and a (local) optimum for local averaging is also an optimum for global VI.

**DP with local averaging**   Assuming $t_j, j = 1, \ldots, M$ are exponential family factors, in the client update at step 5 in Algorithm 2, we can write

$$\Delta t_m^{(s)}(\theta) = \Delta \lambda_m^* \tag{4.3}$$

$$= \lambda^* - \lambda^{(s-1)} \tag{4.4}$$

$$= \frac{1}{N_m} \sum_{k=1}^{N_m} \lambda_{m_k}^* - \lambda^{(s-1)} \tag{4.5}$$

$$= \frac{1}{N_m} \sum_{k=1}^{N_m} \left( \lambda_{m_k}^* - \lambda^{(s-1)} \right). \tag{4.6}$$

We then have the following for guaranteeing DP with local averaging:

**Theorem 7.** *Assume the change in the model parameters $\|\lambda_{m_k}^* - \lambda^{(s-1)}\|_2 \leq C, k = 1, \ldots, N_m$ for some known constant $C$, where $\lambda_{m_k}^*$ is a proposed solution to Equation 4.1, and $\lambda^{(s-1)}$ is the vector of common initial values. Then releasing $\Delta \hat{\lambda}_m^*$ is $(\varepsilon, \delta)$-DP, with $\delta \in (0, 1)$ s.t. $\varepsilon = \mathbb{O}(\delta, q_{sample} = 1, 1, \mathcal{G}_\sigma)$, when*

$$\Delta \hat{\lambda}_m^* = \frac{1}{N_m} \Big[ \sum_{k=1}^{N_m} \left( \lambda_{m_k}^* - \lambda^{(s-1)} \right) + \xi \Big], \tag{4.7}$$

*where $\xi \sim \mathcal{N}(0, \sigma^2 \cdot I)$.*

*Proof.* See A.6 in the Appendix. □

For quantifying the total privacy for $S$ global updates using local averaging, we immediately have the following:

**Corollary 8.** *A composition of $S$ global updates with local averaging using a norm bound $C$ for clipping is $(\varepsilon, \delta)$-DP, with $\delta \in (0, 1)$ s.t. $\varepsilon = \mathbb{O}(\delta, q_{sample} = 1, S, \mathcal{G}_\sigma)$.*

*Proof.* See A.7 in the Appendix. □

As is clear from Corollary 8, with local averaging we pay a privacy cost for each global update, while the local optimisation steps are free. This the opposite of the DP optimisation result in Theorem 6. As mentioned in Corollary 8, in practice we generally need to guarantee the norm bound in Theorem 7 by clipping the change in the model parameters.[3]

Considering how increasing the local data size affects the DP noise level, we have the following:

**Theorem 9.** *With local averaging, the DP noise standard deviation can be scaled as $\mathcal{O}(\frac{1}{N_m})$, where $N_m$ is the number of local partitions. Therefore, the effect of DP noise will vanish on the local factor level when the local dataset size and the number of local partitions grow.*

*Proof.* See A.8 in the Appendix. □

---

[3]We could also enforce DP (including without exponential family factors) by clipping and adding noise directly to the parameters instead of privatising the change in parameters; the clipping would then enforce the parameters to an $\ell_2$−norm ball of radius $C$ around the origin.

Note that Theorem 9 does not say that the DP noise will necessarily vanish on the global approximation level if one client gets more data and does more local partitions, since the total noise level depends on the factors from all the clients. Looking only at Theorem 9, it would seem like increasing the number of local partitions is always beneficial as it decreases the DP noise effect. However, this is not generally the full picture. Zhang et al. (2013) have shown that under some assumptions, the convergence rate of mean estimators (similar to the one we propose) will deteriorate when the number of partitions increases too much. In effect, having fewer samples from which to estimate each local set of parameters increases the estimator variance, which hurts convergence. We have experimentally confirmed this effect with local averaging (see Figure 5 in the Appendix).

The optimal number of local partitions therefore usually balances the decreasing DP noise level with the increasing estimator variance. However, as we show in Theorem 10, there are important special cases, such as the exponential family, where there is no trade-off since the number of local partitions can be increased without affecting the non-DP posterior.

**Theorem 10.** *Assume the effective prior $p_{\backslash j}(\eta)$, and the likelihood $p(x_j|\eta), j \in \{1, \ldots, M\}$ are in a conjugate exponential family, where $\eta$ are the natural parameters. Then the number of partitions used in local averaging does not affect the non-DP posterior.*

*Proof.* See A.9 in the Appendix. □

Finally, Theorem 11 shows that assuming a trusted aggregator, the global approximation noise level can stay constant when adding clients to the protocol, i.e., increasing the number of clients allows every individual client to add less noise while getting the same global DP guarantees.

**Theorem 11.** *Using local averaging with $M$ clients and a shared number of local partitions $N_j = N \ \forall j$ assume the clients have access to a trusted aggregator. Then for any given privacy parameters $\varepsilon, \delta$, the noise standard deviation added by a single client can be scaled as $\mathcal{O}(\frac{1}{\sqrt{M}})$ while guaranteeing the same privacy level.*

*Proof.* See A.10 in the Appendix. □

### 4.2.2 Virtual PVI clients

Running (non-private) PVI with virtual clients is described Algorithm 3.

The full local factor for client $m$ is now $t_m = \prod_{k=1}^{N_m} t_{m,k}$, which is updated only through the virtual factors, and only the change in the full product is ever communicated to the server. This means that when all the virtual factors for client $m$ are in the same exponential family,[4] the parameters for the full local factor $t_m$ are given by

$$\lambda_m = \sum_{k=1}^{N_m} \lambda_{m,k}, \tag{4.10}$$

where $\lambda_{m,k}$ are the parameters for the $k$th virtual factor, and $\Delta t_m^{(s)}(\theta)$ at step 5 in Algorithm 2 can be written as

$$\Delta \lambda_m^* = \sum_{k=1}^{N_m} (\lambda_{m_k}^* - \lambda^{(s-1)}).$$

With virtual PVI clients without DP, doing both local and global updates synchronously corresponds to a regular non-DP PVI run with a synchronous server and $\sum_{j=1}^{M} N_j$ clients. Therefore, all the regular PVI properties (Ashman et al., 2022) derived with a synchronous server immediately hold for non-DP PVI with added virtual clients. In particular, the local ELBO optimisation in this case is equivalent to a variational KL optimisation, and any optimum of the algorithm is also an optimum for global VI.

---

[4]With DP, having all factors from a single exponential family is required to bound the sensitivity.

---
**Algorithm 3** PVI with virtual clients
---
1: Each client $m = 1, \ldots, M$ partitions it's local data into $N_m$ non-overlapping shards and creates corresponding virtual clients, i.e., separate factors $t_{m,k}$ with parameters $\lambda_{m,k}, k = 1, \ldots, N_m$.
2: **for** $s = 1$ to $S$ **do**
3:  Server chooses a subset $b^{(s)} \subseteq \{1, \ldots, M\}$ of clients according to an update schedule and sends the current global model parameters $\lambda^{(s-1)}$.
4:  Each chosen client $m \in b^{(s)}$ updates its virtual clients by locally simulating a single regular PVI update (steps 3-4 in Algorithm 1) with synchronous update schedule $b_m^{(s)} = \{1, \ldots, N_m\}$. The optimised parameters for the $k$th virtual client are given by

$$\lambda_{m_k}^* = \arg\max_{q \in \mathcal{Q}} \left[ \mathbb{E}_q[\log p(x_{m,k}|\theta)] - D_{\text{KL}}(q(\theta|\lambda) \| p_{\backslash m,k}^{(s-1)}(\theta)) \right], \quad k = 1, \ldots, N_m, \tag{4.8}$$

where $p_{\backslash m,k}^{(s-1)}$ is the cavity distribution:

$$p_{\backslash m,k}^{(s-1)}(\theta) \propto p(\theta) \prod_{\substack{j=1 \\ j \neq m}}^{M} t_j(\theta|\lambda_j^{(s-1)}) \prod_{\substack{k'=1 \\ k' \neq k}}^{N_m} t_{m,k'}(\theta|\lambda_{m,k'}^{(s-1)}). \tag{4.9}$$

5:  Each chosen client $m$ updates the local factor and sends the update $\Delta t_m^{(s)}(\theta)$, defined as

$$\Delta t_m^{(s)}(\theta) \propto \frac{t_m(\theta|\lambda_m^*)}{t_m(\theta|\lambda_m^{(s-1)})} \propto \prod_{k=1}^{N_m} \frac{q(\theta|\lambda_{m_k}^*)}{q(\theta|\lambda^{(s-1)})},$$

to the server.
6:  Server updates the global model by incorporating the updated local factors:

$$q(\theta|\lambda^{(s)}) \propto q(\theta|\lambda^{(s-1)}) \prod_{m \in b^{(s)}} \Delta t_m^{(s)}(\theta).$$

7: **end for**
8: **return** Final variational approximation $q(\theta|\lambda^{(S)})$.

---

**DP with virtual PVI clients**  For ensuring DP with virtual clients, again via noising the change in the model parameters as in Section 4.2.1, we have:

**Theorem 12.** *Assume the change in the model parameters $\|\lambda_{m_k}^* - \lambda^{(s-1)}\|_2 \leq C, k = 1, \ldots, N_m$ for some known constant $C$, where $\lambda_{m_k}^*$ is a proposed solution to Equation 4.8, and $\lambda^{(s-1)}$ is the vector of common initial values. Then releasing $\Delta\tilde{\lambda}_m^*$ is $(\varepsilon, \delta)$-DP, with $\delta \in (0, 1)$ s.t. $\varepsilon = \mathbb{O}(\delta, q_{sample} = 1, 1, \mathcal{G}_\sigma)$, when*

$$\Delta\tilde{\lambda}_m^* = \sum_{k=1}^{N_m} \left( \lambda_{m_k}^* - \lambda^{(s-1)} \right) + \xi, \tag{4.11}$$

*where $\xi \sim \mathcal{N}(0, \sigma^2 \cdot I)$.*

*Proof.* See A.11 in the Appendix. □

As an immediate result, Corollary 13 quantifies the total privacy when doing $S$ global updates using virtual PVI clients:

**Corollary 13.** *A composition of $S$ global updates with virtual PVI clients using a norm bound $C$ for clipping is $(\varepsilon, \delta)$-DP, with $\delta \in (0, 1)$ s.t. $\varepsilon = \mathbb{O}(\delta, q_{sample} = 1, S, \mathcal{G}_\sigma)$.*

*Proof.* See A.12 in the Appendix. □

As with local averaging in Corollary 8, and contrasting with DP optimisation in Theorem 6, using Corollary 13 we pay a privacy cost for each global update, but the local optimisation steps are free. And as with local averaging, we usually need to guarantee the assumed norm bound by clipping.

Note that unlike with local averaging in Theorem 9, the noise variance in Equation 4.11 will stay constant with increasing number of local partitions $N_m$. Increasing the number of partitions will decrease the relative effect of the noise if it increases the non-DP sum.

Assuming access to a trusted aggregator, Theorem 14 is a counterpart to Theorem 11 with local averaging: again, the global approximation noise level can stay constant when adding clients to the protocol, meaning that each individual client needs to add less noise while maintaining the same global DP guarantees. The main difference is that with virtual PVI clients each client can choose the number of local partitions freely.

**Theorem 14.** *Assume there are $M$ real clients adding virtual clients, and access to a trusted aggregator. Then for any given privacy parameters $\varepsilon, \delta$, the noise standard deviation added by a single client can be scaled as $\mathcal{O}(\frac{1}{\sqrt{M}})$ while guaranteeing the same privacy level.*

*Proof.* See proof A.13 in the Appendix. □

## 4.3 Summary of technical contributions

We have presented three different implementations of DP-PVI: *DP optimisation*, that is based on perturbing the local optimisation in Section 4.1, as well as *local averaging* in Section 4.2.1 and adding *virtual PVI clients* in Section 4.2.2, which are both based on perturbing the global model updates.

The main idea in DP optimisation is to replace the non-DP optimisation procedure by a DP variant, our main choice being DP-SGD. Hence, DP optimisation inherits all the properties of standard DP-SGD, such as utility guarantees with convex and strongly convex losses. In the more general case of non-convex losses, DP optimisation has no known utility guarantees. Considering the privacy guarantees, with DP optimisation each client enforces DP independently (see Theorem 6), while the global model guarantees result from parallel composition.

In contrast, local averaging and virtual PVI clients are both based on the general idea of adding local data partitioning to mitigate the utility loss from DP noise: each client trains several models on disjoint local data shards, and then combines them for a single global update.

We first showed that local averaging does not fundamentally break the general properties of PVI (see Appendix A): the local ELBO optimisation can be interpreted as a variational KL optimisation, and an optimum of the local averaging algorithm is an optimum for global VI. Privacy for local averaging can be guaranteed by clipping and noising (the change in) the local parameters (see Theorem 7). We showed that the local average under DP approaches the non-DP average on the local factor level when the number of local data shards increases (Theorem 9), and that in the special case of conjugate-exponential family there is no price for increasing the number of local data shards (Theorem 10). Finally, we showed that given access to a suitable secure primitive, we can leverage the other clients to guarantee DP jointly, thereby reducing the amount of noise added by each client while keeping the global model guarantees unchanged (Theorem 11).

Adding virtual PVI clients without DP inherits the properties of non-DP PVI with synchronous updates (e.g., the local ELBO optimisation can be interpreted as a variational KL optimisation, and an optimum of the virtual PVI clients algorithm is an optimum for global VI). We can guarantee privacy via clipping and noising (the change in) the local parameters (Theorem 12). We also showed that, as with local averaging, assuming a suitable secure primitive and guaranteeing DP jointly, the amount of noise added by each client can be reduced while keeping the global model guarantees unchanged (Theorem 14).

Next, we test our proposed methods in Section 5 under various settings to see how they perform in practice.

## 5   Experiments

In this Section we empirically test our proposed methods using logistic regression and Bayesian neural network (BNN) models. Our code for running all the experiments is openly available from `https://github.com/DPBayes/DPPVI`.

We utilise a mean-field Gaussian variational approximation in all experiments. For datasets and prediction tasks, we employ UCI Adult (Kohavi, 1996; Dua & Graff, 2017) predicting whether an individual has income $> 50k$, as well as balanced MIMIC-III health data (Johnson et al., 2016b;a; Goldberger et al., 2000) with an in-hospital mortality prediction task (Harutyunyan et al., 2019).

An important and common challenge in federated learning is that the clients can have very different amounts of data, which cannot usually be assumed to be i.i.d. between the clients. We test the robustness of our approaches to such differences in the data held by each client by using two unbalanced data splits besides the balanced split. In the balanced case, the data is split evenly between all the clients. In both unbalanced data cases half of the clients have a significantly smaller share of data than the larger clients. In the first alternative the small clients only have a few minority class examples, while in the second one they have a considerably larger fraction than the large clients. We defer the details of the data splitting to Appendix B.

In all experiments, we use sequential PVI when not assuming a trusted aggregator, and synchronous PVI otherwise. The number of communications is measured as the number of server-client message exchanges performed by all clients. The actual wall-clock times would depend on the method and implementation: with sequential PVI only one client can update at any one time but communications do not need encryption, while with synchronous PVI all clients can update at the same time but the trusted aggregator methods would also need to account for the time taken by the secure primitive in question. All privacy bounds are calculated numerically with the Fourier accountant (Koskela et al., 2020).[5] The reported privacy budgets include only the privacy cost of the actual learning, while we ignore the privacy leakage due to hyperparameter tuning. More details on the experiments can be found in Appendix B.

We use two baselines: i) DP Bayesian committee machines (BCM with same and split prior, Tresp 2000; Ashman et al. 2022), which are trained with DP-SGD and use a single global communication round to aggregate DP results from all clients, and ii) a centralised DP-VI, which can be trained in our setting with DP-SGD when we assume a trusted aggregator without any communication limits (global VI, trusted aggr., Jälkö et al. 2017).

To measure performance, we report prediction accuracy on held-out data, as well as model log-likelihood as we are interested in uncertainty quantification. Model likelihood effectively measures how well the model can fit unseen data and whether it knows where it is likely to be wrong.

**Logistic regression**   The logistic regression model likelihood is given by

$$p(y|\theta, x) = \sigma(\tilde{x}^\top \theta)^y (1 - \sigma(\tilde{x}^\top \theta))^{1-y},$$

where $y \in \{0, 1\}, \sigma(z) = \frac{1}{1+e^{-z}}$ is the sigmoid function, and $\tilde{x} = [1, x^\top]^\top$ is the augmented data vector with a bias term. In the experiments, we use a standard normal prior for the weights $p(\theta) = \mathcal{N}(\theta|0, I)$ and Monte Carlo (MC) approximate the posterior predictive distribution.

Figures 1 & 2 show the results for logistic regression on Adult and balanced MIMIC-III datasets, respectively, for the three different data splits for 10 clients.

Global VI (global VI, trusted aggr.) is a very strong model utility baseline, that is approximately the best we could hope for. However, as is evident from the results, achieving this baseline requires a trusted aggregator and it uses orders of magnitude more communications than the DP-PVI implementations.[6]

The minimal single-round communication baselines are given by the two BCM variants (BCM same, BCM split). While they are very communication-efficient, the utility varies markedly between the different

---

[5]Available from `https://github.com/DPBayes/PLD-Accountant/`.
[6]Note that the results for global VI do not depend on the data split and hence this baseline curve is the same for all the splits.

settings and they are outperformed by the DP-PVI methods (DP optimisation, local avg, virtual) in several experiments.

For DP-PVI methods, we report results separately without a trusted aggregator (DP optimisation, local avg, virtual) and with a trusted aggregator (local avg, trusted aggr.; virtual, trusted aggr.). In terms of communications, all DP-PVI methods are on par with each other: requiring around an order of magnitude more communications than the minimal communication BCM baselines, and two orders less than the strong utility global VI baseline that always assumes a trusted aggregator.

Our DP optimisation method performs overall well in terms of model utility, always performing slightly worse than the strong utility global VI baseline (global VI, trusted aggr.), being on par with DP-PVI using virtual clients (virtual), and regularly outperforming the BCM baselines (BCM same, BCM split) as well as DP-PVI with local averaging (local avg).

The performance of our local averaging method without access to a trusted aggregator (local avg) seems unstable: it lags behind our other two methods (DP optimisation, virtual) as well as the minimal communication BCM methods (especially BCM split) in several experiments. Given access to a trusted aggregator (local avg, trusted aggr.) the performance improves, as can be expected from the noise scaling in Theorem 11, in several settings significantly so. While it now outperforms our DP optimisation (which does not benefit from the trusted aggregator) and the BCM baselines in many settings, it still regularly lags behind our virtual PVI clients with a trusted aggregator (virtual, trusted aggr.).

Our virtual PVI clients with no trusted aggregator (virtual) performs consistently well in terms of model utility: it lags somewhat behind the strong utility baseline (global VI, trusted aggr.), is on par with our DP optimisation, and outperforms both our local averaging (local avg) and the BCM baselines (BCM same, BCM split) in several experiments. With a trusted aggregator (virtual, trusted aggr.) the model utility improves in line with the noise scaling in Theorem 14, sometimes by a clear margin and even reaching the strong utility baseline (global VI, trusted aggr.) in several settings.

While the results for our local averaging and virtual PVI clients in Figures 1& 2 improve given access to a trusted aggregator even with a very limited communication budget (compare local avg vs. local avg, trusted aggr. and virtual vs. virtual, trusted aggr.), the benefits of having the DP noise scale according to Theorems 11& 14 become more marked with less local data and more clients. Figure 3 shows the results for Adult data with 200 clients: our DP optimisation, which does not benefit from access to a trusted aggregator, now performs clearly worse than local averaging (local avg, trusted aggr.) or virtual PVI clients (virtual, trusted aggr.) which use a trusted aggregator. In contrast to the results in Figures 1& 2, in the more demanding setting in Figure 3, all of our DP-PVI methods clearly outperform the minimum communication BCM baselines (BCM same, BCM split) in model utility.

**BNNs** We expect local averaging and virtual PVI clients to work best when the model has a single, well-defined optimum, since then the local data partitioning should not have a major effect on the resulting model. To test the methods with a more complex model, we use a BNN with one fully connected hidden layer of 50 hidden units, and ReLU non-linearities. Writing $f_\theta(x)$ for the output from the network with parameters $\theta$ and input $x \in \mathbb{R}^d$, the likelihood for $y \in \{0, 1\}$ is a Bernoulli distribution $p(y|\theta, x) = Bern(y|\pi)$, where the class probability $\pi = \sigma^{-1}(f_\theta(x))$, i.e., the logit is predicted by the network. We use a standard normal prior on the weights $p(\theta) = \mathcal{N}(\theta|0, I)$, and MC approximate the predictive distribution:

$$p(y^* = 1|x^*, x, y) \simeq \frac{1}{n_{MC}} \sum_{i=1}^{n_{MC}} p(y^* = 1|\theta^{(i)}, x^*), \theta^{(i)} \sim q(\theta) \ \forall i.$$

The results are shown in Figure 4 for Adult data and 10 clients. The minimum communication baselines (BCM same, BCM split) are very poor in terms of model utility: to achieve good utility with the BNN model in this setting requires some communication between the clients (compare to BCM same and BCM split results in Figure 1 using a tighter privacy budget but simpler logistic regression model). Our DP optimisation works very well in terms of model utility, being almost on par with the high-utility baseline that uses a trusted aggregator (global VI, trusted aggr.). In contrast, both our local averaging (local avg) and virtual

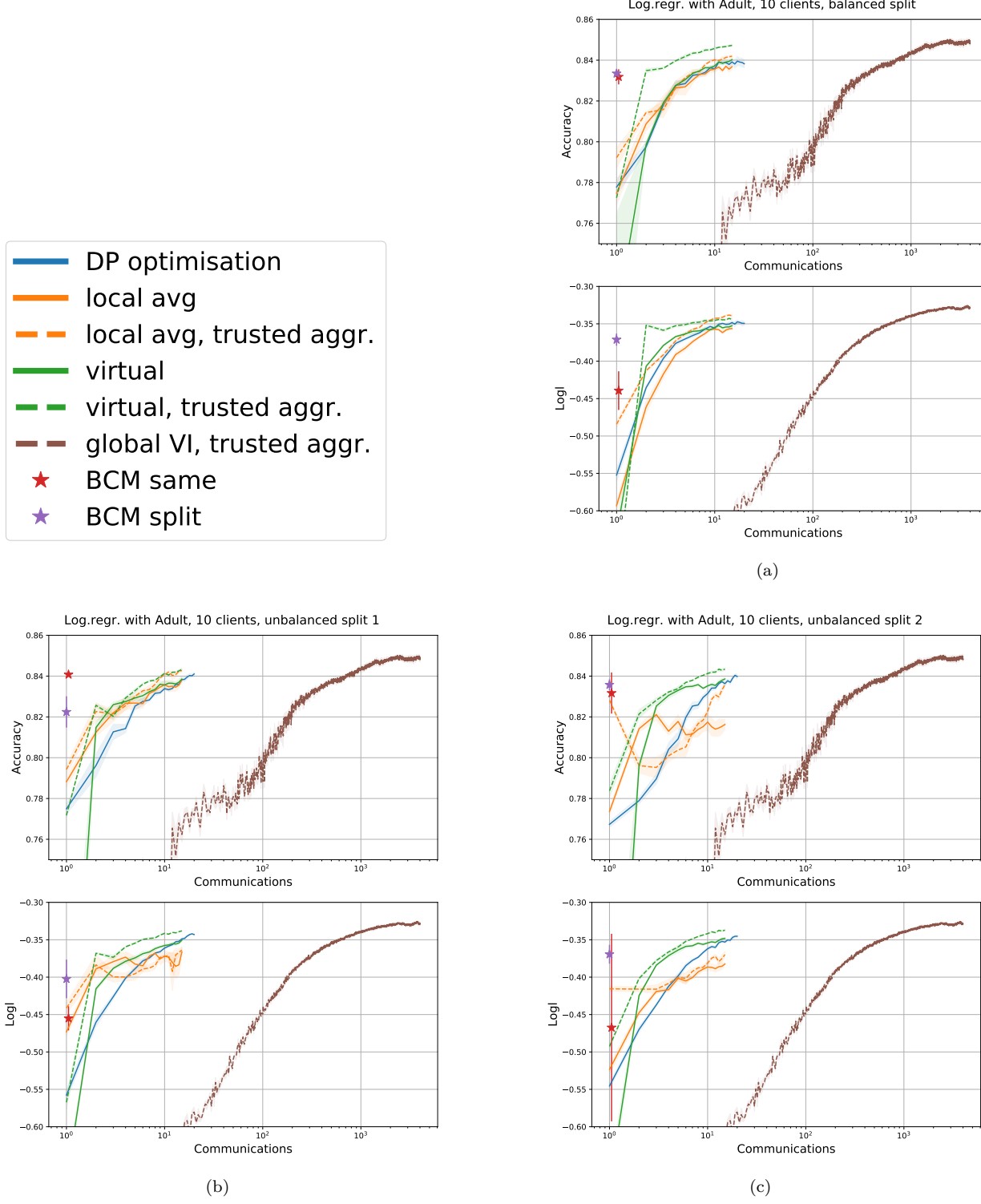

Figure 1: $(1, 10^{-5})$-DP logistic regression, Adult data with 10 clients: mean over 5 seeds with SEM. a) balanced split, b) unbalanced split 1, c) unbalanced split 2.

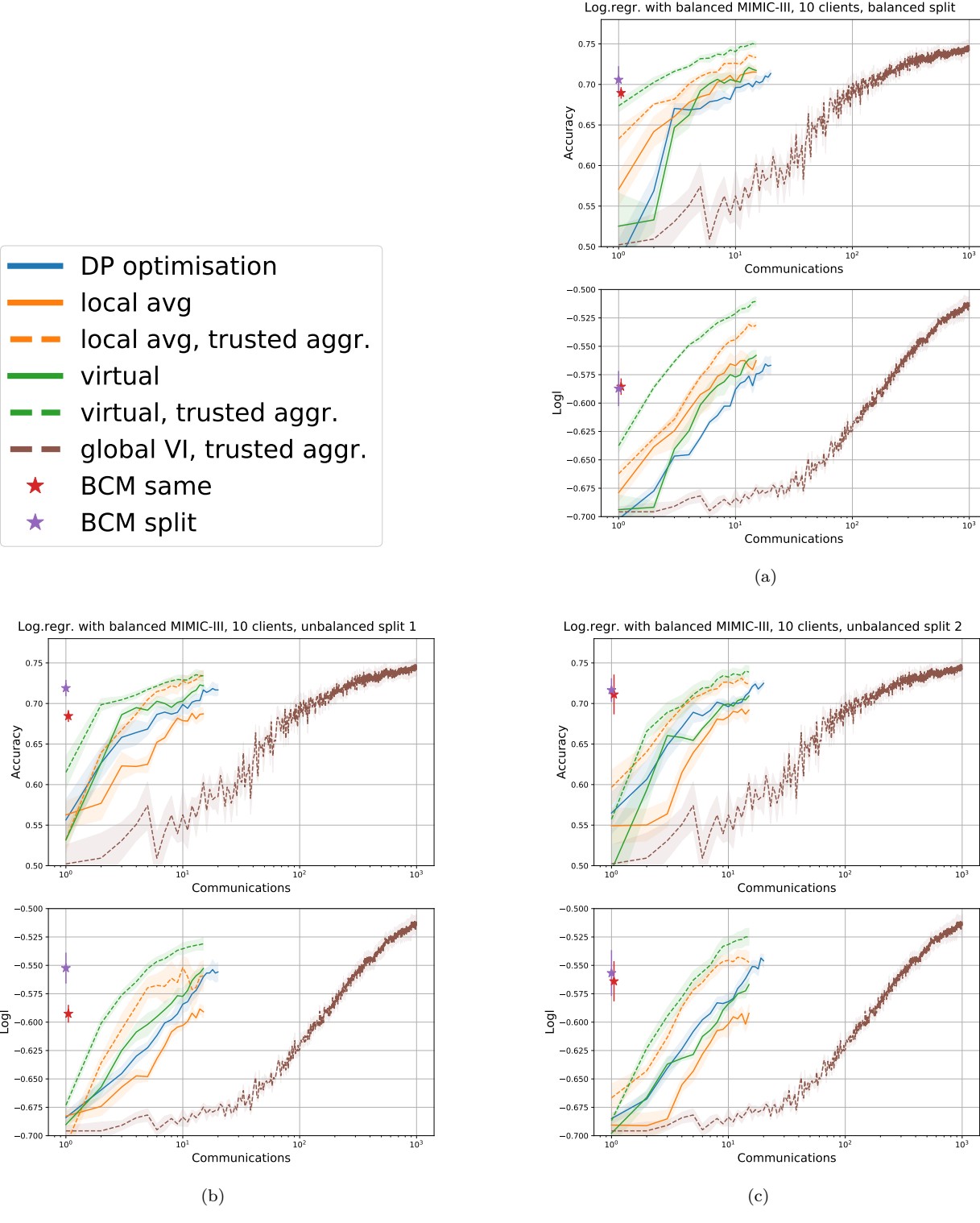

Figure 2: $(1, 10^{-5})$-DP logistic regression, balanced MIMIC-III data with 10 clients: mean over 5 seeds with SEM. a) balanced split, b) unbalanced split 1, c) unbalanced split 2.

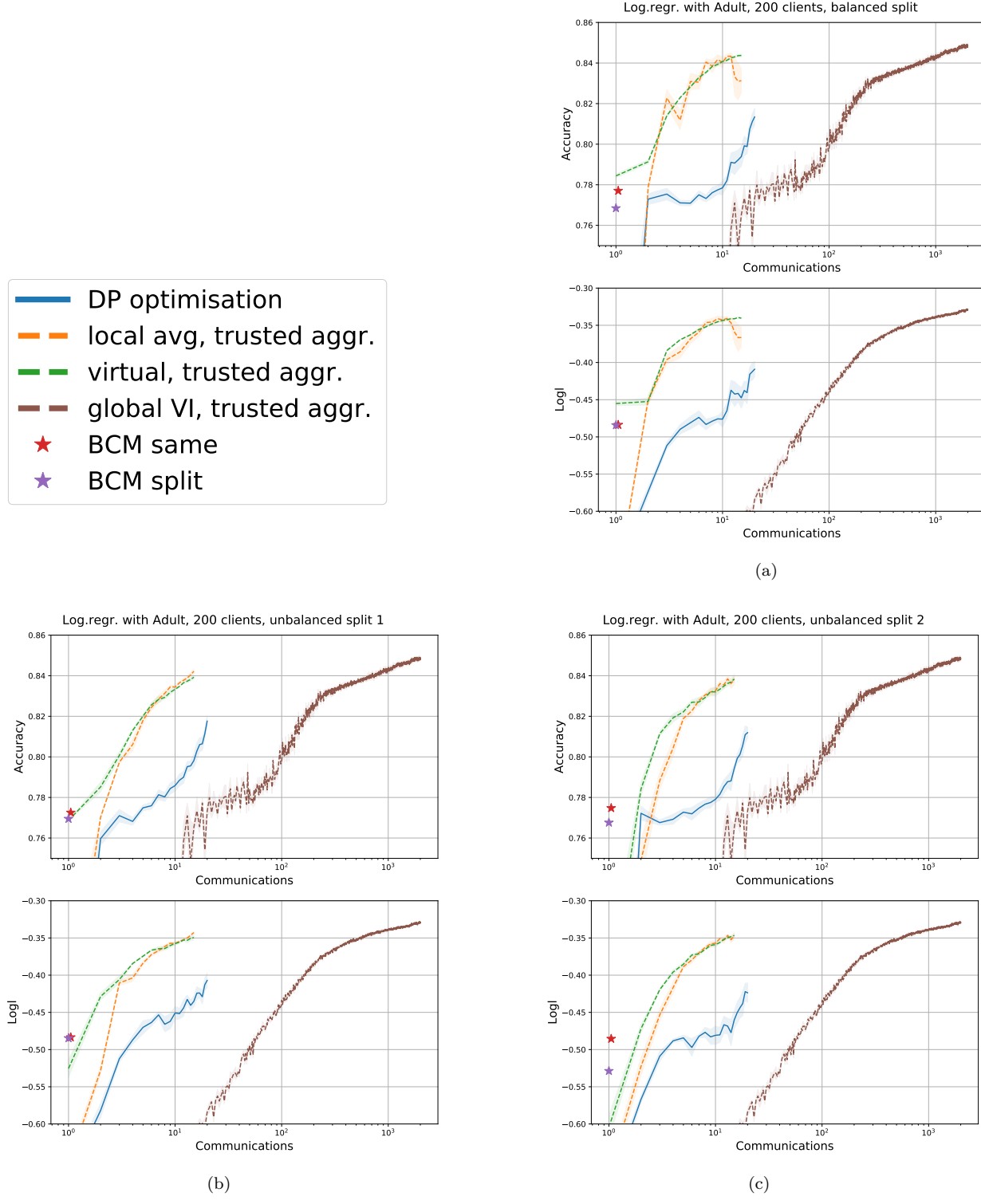

Figure 3: $\left(\frac{1}{2}, 10^{-5}\right)$-DP logistic regression with Adult data and 200 clients: mean over 5 seeds with SEM. a) balanced split, b) unbalanced split 1, c) unbalanced split 2.

PVI clients (virtual) perform clearly worse. They are also much more likely to diverge and seem to require generally more careful tuning of the hyperparameters to reach any reasonable performance.

## 6 Discussion

We have proposed three different implementations of DP-PVI, one based on perturbing the local optimisation (DP optimisation), and two based on perturbing the model updates (local averaging and virtual PVI clients). As is clear from the empirical results in Section 5, no single method dominates the others in all settings. Instead, the method needs to be chosen according to the problem at hand. However, we can derive guidelines for choosing the most suitable method based on the theoretical results as well as on the empirical experiments.

DP optimisation is a good candidate method with simple or complex models regardless of the number of clients as long as the clients have enough local data. However, with little local data and more clients, since it cannot easily benefit from secure primitives, the performance can be sub-optimal.

In contrast, local averaging and virtual PVI clients work best with relatively simple models, while the performance with more complex models can easily lag behind DP optimisation results. However, given access to a trusted aggregator both methods can leverage other clients to reduce the amount of DP noise required. Based on our experiments, from the two methods local averaging is harder to tune properly and can be unstable whereas using virtual PVI clients is a more robust alternative.

### Acknowledgments

The authors would like to thank Mrinank Sharma, Michael Hutchinson, Stratis Markou and Antti Koskela for useful discussions. The authors acknowledge CSC – IT Center for Science, Finland, and the Finnish Computing Competence Infrastructure (FCCI) for computational and data storage resources.

Mikko Heikkilä was at the University of Helsinki during this work.

Matthew Ashman is supported by the George and Lilian Schiff Foundation.

Siddharth Swaroop was at the University of Cambridge during this work, where he was supported by an EPSRC DTP studentship and a Microsoft Research EMEA PhD Award.

Richard E. Turner is supported by Google, Amazon, ARM, Improbable and EPSRC grant EP/T005386/1.

This work was supported by the Academy of Finland (Flagship programme: Finnish Center for Artificial Intelligence, FCAI; and grant 325573), the Strategic Research Council at the Academy of Finland (Grant 336032) as well as the European Union (Project 101070617). Views and opinions expressed are however those of the author(s) only and do not necessarily reflect those of the European Union or the European Commission. Neither the European Union nor the granting authority can be held responsible for them.

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

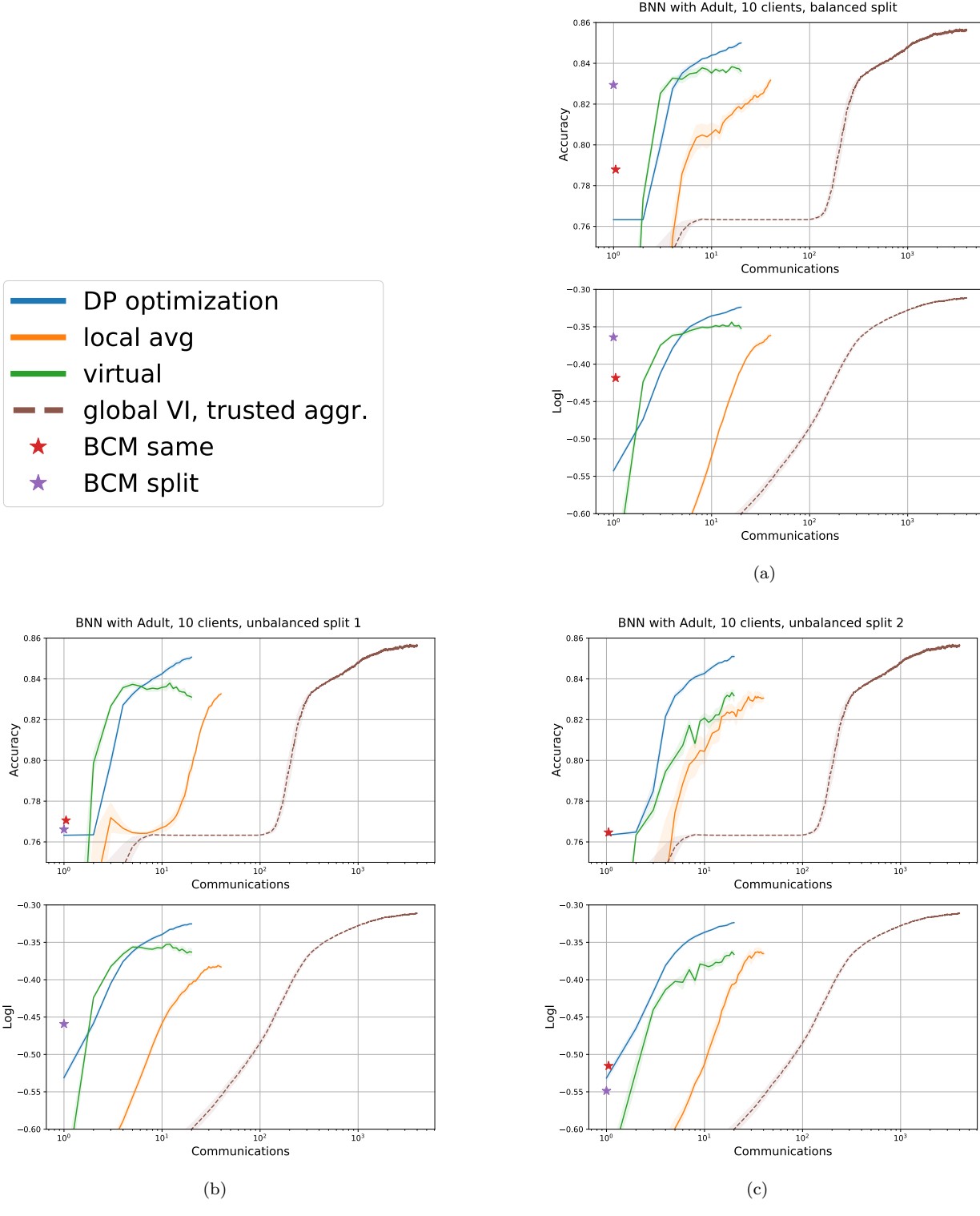

Figure 4: $(2, 10^{-5})$-DP 1-layer BNN with Adult data and 10 clients: mean over 5 seeds with SEM. a) balanced split, b) unbalanced split 1, c) unbalanced split 2.

Raef Bassily, Adam Smith, and Abhradeep Thakurta. Private empirical risk minimization: Efficient algorithms and tight error bounds. In *Proceedings of the 2014 IEEE 55th Annual Symposium on Foundations of Computer Science*, FOCS '14, pp. 464–473, Washington, DC, USA, 2014. IEEE Computer Society. ISBN 978-1-4799-6517-5. doi: 10.1109/FOCS.2014.56. URL http://dx.doi.org/10.1109/FOCS.2014.56.

José M Bernardo and Adrian F M Smith. *Bayesian Theory*. John Wiley & Sons, Inc., 1994.

Garrett Bernstein and Daniel Sheldon. Differentially private bayesian inference for exponential families. In S. Bengio and H. Wallach and H. Larochelle and K. Grauman and N. Cesa-Bianchi and R. Garnett (ed.), *Advances in Neural Information Processing Systems*, volume 31, 2018.

H Brendan McMahan, Eider Moore, Daniel Ramage, Seth Hampson, and Blaise Agüera y Arcas. Communication-Efficient learning of deep networks from decentralized data. February 2016.

Tamara Broderick, Nicholas Boyd, Andre Wibisono, Ashia C. Wilson, and Michael I. Jordan. Streaming variational Bayes. In *Advances in Neural Information Processing Systems*, 2013.

Thang D Bui, Cuong V Nguyen, Siddharth Swaroop, and Richard E Turner. Partitioned variational inference: A unified framework encompassing federated and continual learning. November 2018.

David L Chaum. Untraceable electronic mail, return addresses, and digital pseudonyms. *Commun. ACM*, 24 (2):84–90, February 1981.

Wei-Ning Chen, Christopher A Choquette-Choo, Peter Kairouz, and Ananda Theertha Suresh. The fundamental price of secure aggregation in differentially private federated learning. March 2022.

Albert Cheu, Adam Smith, Jonathan Ullman, David Zeber, and Maxim Zhilyaev. Distributed differential privacy via shuffling. In *Annual International Conference on the Theory and Applications of Cryptographic Techniques*, pp. 375–403. Springer, 2019.

Christos Dimitrakakis, Blaine Nelson, Aikaterini Mitrokotsa, and Benjamin I. P. Rubinstein. Robust and private Bayesian inference. In *Proc. ALT 2014*, pp. 291–305. 2014.

Christos Dimitrakakis, Blaine Nelson, Zuhe Zhang, Aikaterini Mitrokotsa, and Benjamin I. P. Rubinstein. Differential privacy for Bayesian inference through posterior sampling. *Journal of Machine Learning Research*, 18(11):1–39, 2017.

Dheeru Dua and Casey Graff. UCI machine learning repository, 2017. URL http://archive.ics.uci.edu/ml.

Cynthia Dwork and Aaron Roth. The algorithmic foundations of differential privacy. *Found. Trends Theor. Comput. Sci.*, 9(3–4):211–407, August 2014. ISSN 1551-305X. doi: 10.1561/0400000042. URL http://dx.doi.org/10.1561/0400000042.

Cynthia Dwork and Adam Smith. Differential privacy for statistics: What we know and what we want to learn. *Journal of Privacy and Confidentiality*, 1(2), Apr. 2010. doi: 10.29012/jpc.v1i2.570. URL https://journalprivacyconfidentiality.org/index.php/jpc/article/view/570.

Cynthia Dwork, Krishnaram Kenthapadi, Frank McSherry, Ilya Mironov, and Moni Naor. Our data, ourselves: Privacy via distributed noise generation. In *Annual International Conference on the Theory and Applications of Cryptographic Techniques*, pp. 486–503. Springer, 2006a.

Cynthia Dwork, Frank McSherry, Kobbi Nissim, and Adam Smith. Calibrating noise to sensitivity in private data analysis. In *Proc. TCC 2006*, pp. 265–284. 2006b. ISBN 978-3-540-32732-5. doi: 10.1007/11681878_14. URL http://dx.doi.org/10.1007/11681878_14.

James Foulds, Joseph Geumlek, Max Welling, and Kamalika Chaudhuri. On the theory and practice of privacy-preserving Bayesian data analysis. In *Proc. 32nd Conf. on Uncertainty in Artificial Intelligence (UAI 2016)*, 2016.

James R. Foulds, Mijung Park, Kamalika Chaudhuri, and Max Welling. Variational Bayes in private settings (VIPS) (extended abstract). In Christian Bessiere (ed.), *Proceedings of the Twenty-Ninth International Joint Conference on Artificial Intelligence, IJCAI-20*, pp. 5050–5054. International Joint Conferences on Artificial Intelligence Organization, 7 2020. doi: 10.24963/ijcai.2020/705. URL `https://doi.org/10.24963/ijcai.2020/705`. Journal track.

Joseph Geumlek, Shuang Song, and Kamalika Chaudhuri. Rényi differential privacy mechanisms for posterior sampling. In *Proceedings of the 31st International Conference on Neural Information Processing Systems*, NIPS'17, pp. 5295–5304, USA, 2017. Curran Associates Inc.

Zoubin Ghahramani and H. Attias. Online variational Bayesian learning. In *NIPS Workshop on Online Learning*, 2000.

A. Goldberger, L. Amaral, L. Glass, J. Hausdorff, P. C. Ivanov, R. Mark, J.E. Mietus, G.B. Moody, C.K Peng, and H. E. Stanley. PhysioBank, PhysioToolkit, and PhysioNet: Components of a new research resource for complex physiologic signals. *Circulation*, 101(23), 2000.

Hrayr Harutyunyan, Hrant Khachatrian, David C Kale, Greg Ver Steeg, and Aram Galstyan. Multitask learning and benchmarking with clinical time series data. *Scientific Data*, 6(96), June 2019.

Mikko Heikkilä, Eemil Lagerspetz, Samuel Kaski, Kana Shimizu, Sasu Tarkoma, and Antti Honkela. Differentially private bayesian learning on distributed data. In I. Guyon, U. Von Luxburg, S. Bengio, H. Wallach, R. Fergus, S. Vishwanathan, and R. Garnett (eds.), *Advances in Neural Information Processing Systems*, volume 30. Curran Associates, Inc., 2017. URL `https://proceedings.neurips.cc/paper/2017/file/dfce06801e1a85d6d06f1fdd4475dacd-Paper.pdf`.

Mikko Heikkilä, Joonas Jälkö, Onur Dikmen, and Antti Honkela. Differentially private markov chain monte carlo. In H. Wallach, H. Larochelle, A. Beygelzimer, F. d'Alché-Buc, E. Fox, and R. Garnett (eds.), *Advances in Neural Information Processing Systems*, volume 32. Curran Associates, Inc., 2019. URL `https://proceedings.neurips.cc/paper/2019/file/074177d3eb6371e32c16c55a3b8f706b-Paper.pdf`.

Geoffrey E Hinton and Drew Van Camp. Keeping the neural networks simple by minimizing the description length of the weights. In *Conference on Computational Learning Theory*, pp. 5–13, 1993.

Antti Honkela, Mrinal Das, Arttu Nieminen, Onur Dikmen, and Samuel Kaski. Efficient differentially private learning improves drug sensitivity prediction. *Biology Direct*, 13(1):1, 2018.

Joonas Jälkö, Antti Honkela, and Onur Dikmen. Differentially private variational inference for non-conjugate models. In *Proc. UAI 2017*, 2017. URL `http://auai.org/uai2017/proceedings/papers/152.pdf`.

Alistair Johnson, Tom Pollard, and Roger Mark. MIMIC-III clinical database (version 1.4). PhysioNet, 2016a. URL `https://doi.org/10.13026/C2XW26`.

Alistair E W Johnson, Tom J Pollard, Lu Shen, Li-Wei H Lehman, Mengling Feng, Mohammad Ghassemi, Benjamin Moody, Peter Szolovits, Leo Anthony Celi, and Roger G Mark. MIMIC-III, a freely accessible critical care database. *Scientific Data*, 3(160035), May 2016b.

Michael I Jordan, Zoubin Ghahramani, Tommi S Jaakkola, and Lawrence K Saul. An introduction to variational methods for graphical models. *Mach. Learn.*, 37(2):183–233, 1999.

Peter Kairouz, H. Brendan McMahan, Brendan Avent, Aurélien Bellet, Mehdi Bennis, Arjun Nitin Bhagoji, Kallista Bonawitz, Zachary Charles, Graham Cormode, Rachel Cummings, Rafael G. L. D'Oliveira, Hubert Eichner, Salim El Rouayheb, David Evans, Josh Gardner, Zachary Garrett, Adrià Gascón, Badih Ghazi, Phillip B. Gibbons, Marco Gruteser, Zaid Harchaoui, Chaoyang He, Lie He, Zhouyuan Huo, Ben Hutchinson, Justin Hsu, Martin Jaggi, Tara Javidi, Gauri Joshi, Mikhail Khodak, Jakub Konečný, Aleksandra Korolova, Farinaz Koushanfar, Sanmi Koyejo, Tancrède Lepoint, Yang Liu, Prateek Mittal, Mehryar Mohri, Richard Nock, Ayfer Özgür, Rasmus Pagh, Mariana Raykova, Hang Qi, Daniel Ramage, Ramesh Raskar, Dawn

Song, Weikang Song, Sebastian U. Stich, Ziteng Sun, Ananda Theertha Suresh, Florian Tramèr, Praneeth Vepakomma, Jianyu Wang, Li Xiong, Zheng Xu, Qiang Yang, Felix X. Yu, Han Yu, and Sen Zhao. Advances and open problems in federated learning. 2019. doi: 10.48550/ARXIV.1912.04977. URL https://arxiv.org/abs/1912.04977.

Mohammad Emtiyaz Khan and Wu Lin. Conjugate-computation variational inference : Converting variational inference in non-conjugate models to inferences in conjugate models. In *International Conference on Artificial Intelligence and Statistics*, 2017.

Diederik P. Kingma and Jimmy Ba. Adam: A method for stochastic optimization, 2014. URL https://arxiv.org/abs/1412.6980.

David A. Knowles and Tom Minka. Non-conjugate variational message passing for multinomial and binary regression. In *Advances in Neural Information Processing Systems*, pp. 1701–1709, 2011.

Ron Kohavi. Scaling up the accuracy of naive-bayes classifiers: A decision-tree hybrid. In *Proceedings of the Second International Conference on Knowledge Discovery and Data Mining*, KDD'96, pp. 202–207. AAAI Press, 1996.

Antti Koskela, Joonas Jälkö, and Antti Honkela. Computing tight differential privacy guarantees using FFT. In *International Conference on Artificial Intelligence and Statistics*, pp. 2560–2569. PMLR, 2020.

Bai Li, Changyou Chen, Hao Liu, and Lawrence Carin. On connecting stochastic gradient mcmc and differential privacy. In Kamalika Chaudhuri and Masashi Sugiyama (eds.), *Proceedings of the Twenty-Second International Conference on Artificial Intelligence and Statistics*, volume 89 of *Proceedings of Machine Learning Research*, pp. 557–566. PMLR, 16–18 Apr 2019. URL https://proceedings.mlr.press/v89/li19a.html.

Yingzhen Li, José Miguel Hernández-Lobato, and Richard E Turner. Stochastic expectation propagation. In C. Cortes, N. Lawrence, D. Lee, M. Sugiyama, and R. Garnett (eds.), *Advances in Neural Information Processing Systems*, volume 28. Curran Associates, Inc., 2015. URL https://proceedings.neurips.cc/paper/2015/file/f3bd5ad57c8389a8a1a541a76be463bf-Paper.pdf.

Tom Minka. Divergence measures and message passing. Technical Report MSR-TR-2005-173, January 2005. URL https://www.microsoft.com/en-us/research/publication/divergence-measures-and-message-passing/.

Cuong V. Nguyen, Yingzhen Li, Thang D. Bui, and Richard E. Turner. Variational continual learning. In *International Conference on Learning Representations*, 2018.

Kobbi Nissim, Sofya Raskhodnikova, and Adam Smith. Smooth sensitivity and sampling in private data analysis. *Proceedings of the thirty-ninth annual ACM symposium on Theory of computing - STOC '07*, pp. 75, 2007.

Vibhor Rastogi and Suman Nath. Differentially private aggregation of distributed time-series with transformation and encryption. In *Proceedings of the 2010 ACM SIGMOD International Conference on Management of Data*, SIGMOD '10, pp. 735–746, New York, NY, USA, 2010. ACM.

Ossi Räisä, Antti Koskela, and Antti Honkela. Differentially private Hamiltonian Monte Carlo, 2021. URL https://arxiv.org/abs/2106.09376.

Adi Shamir. How to share a secret. *Commun. ACM*, 22(11):612–613, November 1979.

Mrinank Sharma, Michael Hutchinson, Siddharth Swaroop, Antti Honkela, and Richard E. Turner. Differentially private federated variational inference, 2019. URL https://arxiv.org/abs/1911.10563.

Shuang Song, Kamalika Chaudhuri, and Anand D. Sarwate. Stochastic gradient descent with differentially private updates. In *Proc. GlobalSIP 2013*, pp. 245–248, 2013. doi: 10.1109/GlobalSIP.2013.6736861. URL https://doi.org/10.1109/GlobalSIP.2013.6736861.

Volker Tresp. A Bayesian committee machine. *Neural Computation*, 12(11):2719–2741, 2000.

Margarita Vinaroz and Mijung Park. Differentially private stochastic expectation propagation (DP-SEP), 2021. URL https://arxiv.org/abs/2111.13219.

J Wainwright, M I Jordan, Martin J Wainwright, and Michael I Jordan. Graphical models, exponential families, and variational inference. *Mach. Learn.*, 1:1–2, 2008.

Matt P. Wand. Fully simplified multivariate Normal updates in non-conjugate variational message passing. *Journal of Machine Learning Research*, 15:1351–1369, 2014.

Yu-Xiang Wang, Stephen Fienberg, and Alex Smola. Privacy for free: Posterior sampling and stochastic gradient monte carlo. In Francis Bach and David Blei (eds.), *Proceedings of the 32nd International Conference on Machine Learning*, volume 37 of *Proceedings of Machine Learning Research*, pp. 2493–2502, Lille, France, 07–09 Jul 2015. PMLR. URL https://proceedings.mlr.press/v37/wangg15.html.

Kang Wei, Jun Li, Ming Ding, Chuan Ma, Howard H Yang, Farhad Farokhi, Shi Jin, Tony Q S Quek, and H Vincent Poor. Federated learning with differential privacy: Algorithms and performance analysis. *IEEE Trans. Inf. Forensics Secur.*, 15:3454–3469, 2020.

John Winn, Christopher M. Bishop, and Tommi Jaakkola. Variational message passing. *Journal of Machine Learning Research*, 6:661–694, 2005.

Sinan Yıldırım and Beyza Ermiş. Exact MCMC with differentially private moves. *Statistics and Computing*, 29(5):947–963, sep 2019. ISSN 0960-3174. doi: 10.1007/s11222-018-9847-x. URL https://doi.org/10.1007/s11222-018-9847-x.

Cheng Zhang, Judith Bütepage, Hedvig Kjellström, and Stephan Mandt. Advances in variational inference. *IEEE Transactions on Pattern Analysis and Machine Intelligence*, 41(8):2008–2026, 2019. doi: 10.1109/TPAMI.2018.2889774.

Yuchen Zhang, John C Duchi, and Martin J Wainwright. Communication-efficient algorithms for statistical optimization. *J. Mach. Learn. Res.*, 14:3321–3363, 2013.

Zuhe Zhang, Benjamin Rubinstein, and Christos Dimitrakakis. On the differential privacy of Bayesian inference. In *Proc. Conf. AAAI Artif. Intell. 2016*, 2016.

Ligeng Zhu, Zhijian Liu, and Song Han. Deep leakage from gradients. In H. Wallach, H. Larochelle, A. Beygelzimer, F. d'Alché-Buc, E. Fox, and R. Garnett (eds.), *Advances in Neural Information Processing Systems*, volume 32. Curran Associates, Inc., 2019. URL https://proceedings.neurips.cc/paper/2019/file/60a6c4002cc7b29142def8871531281a-Paper.pdf.

# A   Appendix: theorems and proofs

This Appendix contains all proofs and some additional theorems omitted from the main text. For easy of reading, we state all the theorems before the proofs.

**Privacy via local optimisation: DP optimisation**

**Theorem A.1.** *Running DP-SGD for client-level optimisation in Algorithm 1, using subsampling fraction $q_{sample} \in (0,1]$ on the local data level for $T$ local optimisation steps in total, with $S$ global updates interleaved with the local steps, the resulting model is $(\varepsilon, \delta)$-DP, with $\delta \in (0,1)$ s.t. $\varepsilon = \mathbb{O}(\delta, q_{sample}, T, \mathcal{G}_\sigma)$.*

*Proof.* Standard DP-SGD theory (Song et al., 2013; Bassily et al., 2014; Abadi et al., 2016) ensures that the local optimised approximation is DP after a given number of local optimisation steps by a given client $m$, when on every local step we clip each per-example gradient to enforce a known $\ell_2$-norm bound $C$, add iid Gaussian noise with standard deviation $\sigma$ scaled by $C$ to each dimension, and the privacy amplification by

subsampling factor $q_{sample}$ for the sampling without replacement function (see Definition 4) is calculated from the fraction of local data utilised on each step. Since the global update does not access the sensitive data, the global model is DP w.r.t. data held by client $m$ after the global update due to post-processing guarantees. Hence, when accounting for DP for client $m$, the total number of compositions is $T$, regardless of the number of global updates. The total privacy is therefore $(\varepsilon, \delta)$, when $\delta$ is such that $\varepsilon = \mathbb{O}(\delta, q_{sample}, T, \mathcal{G}_\sigma)$. $\qquad\square$

With Theorem A.1, DP is guaranteed independently by each client w.r.t. their own data, and hence the global model will have DP guarantees w.r.t. any clients' data via parallel composition, i.e., the global model is $(\epsilon_{max}, \delta_{max})$-DP w.r.t. any single training data sample with $\epsilon_{max} = \max\{\epsilon_1, \ldots, \epsilon_M\}, \delta_{max} = \max\{\delta_1, \ldots, \delta_M\}$, where $\epsilon_m, \delta_m$ are the parameters used by client $m$. In all the experiments in this paper, we use a common $(\epsilon, \delta)$ budget shared by all the clients, and sampling without replacement on the local data level as the subsampling method.

**Properties of non-DP local averaging**  The following properties are the local averaging counterparts to the regular PVI properties shown by Ashman et al. (2022). We write $n_{m,k}$ for the number of samples in shard $k$ at client $m$ after the initial partitioning, so $\sum_{k=1}^{N_m} n_{m,k} = n_m$.

***Property A.2*** (cf. Property 2.1 of Ashman et al. 2022). *Maximizing the local ELBO*

$$\mathcal{L}_{m,k}^{(s)}(q(\theta)) := \int d\theta q(\theta) \log \frac{[p(x_{m,k}|\theta)]^{N_m} q^{(s-1)}(\theta)}{q(\theta) t_m^{(s-1)}(\theta)}$$

*is equivalent to the KL optimization*

$$q^{(s)}(\theta) = \arg\min_q D_{\mathrm{KL}}(q(\theta) \| \hat{p}_{m,k}^{(s)}(\theta)),$$

*where $\hat{p}_{m,k}^{(s)}(\theta) = \frac{1}{\hat{Z}_{m,k}^{(s)}} p(\theta) \prod_{j \neq m} t_j^{(s-1)}(\theta) \cdot [p(x_{m,k}|\theta)]^{N_m}$ is the tempered tilted distribution before global update $s$ for local shard $k$ at client $m$.*

*Proof.* The proof is identical to the one in (Ashman et al., 2022, A.1) when we replace the full local likelihood $p(x_m|\theta)$ by the tempered likelihood $[p(x_{m,k}|\theta)]^{N_m}$ for shard $k$. $\qquad\square$

***Property A.3*** (cf. Property 2.2 of Ashman et al. 2022). *Let $q^*(\theta) = p(\theta) \prod_{j=1}^{M} t_j^*(\theta)$ be a fixed point for local averaging, $\mathcal{L}_{m,k}^*(q(\theta)) = \int d\theta q(\theta) \log \frac{q^*(\theta)[p(x_{m,k}|\theta)]^{N_m}}{q(\theta) t_m^*(\theta)}$ local ELBO at the fixed point w.r.t. shard $k$ at client $m$, and $\mathcal{L}(q(\theta)) = \int d\theta q(\theta) \log \frac{p(\theta)p(x|\theta)}{q(\theta)}$ global ELBO. Then*

*1. $\sum_{j=1}^{M} \frac{1}{N_j} \sum_{k=1}^{N_j} \mathcal{L}_{j,k}^*(q^*(\theta)) = \mathcal{L}(q^*(\theta)) - \log Z_{q^*}.$*

*2. If $q^*(\theta) = \arg\max_q \mathcal{L}_{j,k}(q(\theta))$ for all $j, k$, then $q^*(\theta) = \arg\max_q \mathcal{L}(q(\theta))$.*

*Proof.* 1. Directly from the definition we have

$$\mathcal{L}(q^*(\theta)) - \log Z_{q^*} = \int d\theta q^*(\theta) \log \frac{p(\theta)p(x|\theta)}{q^*(\theta)Z_{q^*}} \tag{A.1}$$

$$= \int d\theta q^*(\theta)[\log p(x|\theta) - \log \frac{p(\theta) \prod_{j=1}^{M} t_j^*(\theta)}{p(\theta)}] \tag{A.2}$$

$$= \sum_{j=1}^{M} \int d\theta q^*(\theta)[\log \prod_{k=1}^{N_j} p(x_{j,k}|\theta) - \log t_j^*(\theta)] \tag{A.3}$$

$$= \sum_{j=1}^{M} \int d\theta q^*(\theta)[\frac{1}{N_j} N_j \sum_{k=1}^{N_j} \log p(x_{j,k}|\theta) - \log \frac{q^*(\theta)t_j^*(\theta)}{q^*(\theta)}] \tag{A.4}$$

$$= \sum_{j=1}^{M} \frac{1}{N_j} \sum_{k=1}^{N_j} \int d\theta q^*(\theta)[\log \frac{q^*(\theta)[p(x_{j,k}|\theta)]^{N_j}}{q^*(\theta)t_j^*(\theta)}] \tag{A.5}$$

$$= \sum_{j=1}^{M} \frac{1}{N_j} \sum_{k=1}^{N_j} \mathcal{L}_{j,k}^*(q^*(\theta)). \tag{A.6}$$

Assume $q^*(\theta) = \arg\max_q \mathcal{L}_{j,k}(q(\theta))$ for all $j, k$. Then $q^*(\theta) = \arg\max_q \mathcal{L}_{j,k}^*(q(\theta))$ for all $j, k$ implying that $\frac{d}{d\lambda_q} \mathcal{L}_{j,k}^*(q^*(\theta)) = 0$ for all $j, k$. Furthermore, since $q^*(\theta)$ is a maximizer, the Hessian $\frac{d^2}{d\lambda_q d\lambda_q^T} \mathcal{L}_{j,k}^*(q^*(\theta))$ is negative definite for all $j, k$. Looking at the first part of the proof we can write

$$\frac{d}{d\lambda_q} \mathcal{L}(q^*(\theta)) = \sum_{j=1}^{M} \frac{1}{N_j} \sum_{k=1}^{N_j} \frac{d}{d\lambda_q} \mathcal{L}_{j,k}^*(q^*(\theta)) = 0, \text{ and} \tag{A.7}$$

$$\frac{d^2}{d\lambda_q d\lambda_q^T} \mathcal{L}(q^*(\theta)) = \sum_{j=1}^{M} \frac{1}{N_j} \sum_{k=1}^{N_j} \frac{d^2}{d\lambda_q d\lambda_q^T} \mathcal{L}_{j,k}^*(q^*(\theta)). \tag{A.8}$$

From Equation A.7 we see that $q^*(\theta)$ is a fixed point of $\mathcal{L}(q(\theta))$, and since the Hessian in Equation A.8 can be expressed by summing negative definite matrices and multiplying them by positive numbers, the resulting Hessian is also negative definite, and hence $q^*(\theta)$ maximizes the global ELBO $\mathcal{L}(q(\theta))$.

$\square$

**Property A.4** (cf. Property 3.2 of Ashman et al. 2022)**.** *Assume the prior and approximate likelihood factors are in the unnormalized exponential family $t_m(\theta) = t_m(\theta; \lambda_m) = \exp(\lambda_m^T T(\theta))$, so the variational distribution is in the normalized exponential family $q(\theta) = \exp(\lambda_q^T T(\theta) - A(\lambda_q))$. Then a stationary point of the local ELBO at global update $s$ for the $k$th local model at client $m$, $\frac{d\mathcal{L}_{m,k}^{(s)}(q_k(\theta))}{d\lambda_q} = 0$, implies*

$$\lambda_{m,k}^{(s)} = N_m \frac{d}{d\mu_{q^{(s-1)}}} \mathbb{E}_{q^{(s-1)}}[\log p(x_{m,k}|\theta)].$$

*In addition, a stationary point for all $N_m$ local models' ELBO implies*

$$\lambda_m^{(s)} = \frac{1}{N_m} \sum_{k=1}^{N_m} \lambda_{m,k}^{(s)} = \frac{d}{d\mu_{q^{(s-1)}}} \mathbb{E}_{q^{(s-1)}}[\log p(x_m|\theta)],$$

*which matches the regular PVI fixed point equation.*

*Proof.* Writing $\mathcal{L}_{m,k}^{(s)}(q_k(\theta)) = \int d\theta q_k(\theta) \log \frac{[p(x_{m,k}|\theta)]^{N_m} q^{(s-1)}(\theta)}{q_k(\theta)t_m^{(s-1)}(\theta)}$ and noting that all $N_m$ local models are started in parallel from the same point (so $\mu_{q_k^{(s-1)}} = \mu_{q^{(s-1)}}, q_k^{(s-1)} = q^{(s-1)} \forall k$), then following the proof in

(Ashman et al., 2022, Supplement A.4) with minor changes establishes the first claim:

$$\lambda_{m,k}^{(s)} = N_m \frac{d}{d\mu_{q^{(s-1)}}} \mathbb{E}_{q^{(s-1)}} [\log p(x_{m,k}|\theta)].$$

Looking now at the average of local parameters we have

$$\lambda_m^{(s)} = \frac{1}{N_m} \sum_{k=1}^{N_m} \lambda_{m,k}^{(s)} \tag{A.9}$$

$$= \frac{d}{d\mu_{q^{(s-1)}}} \mathbb{E}_{q^{(s-1)}} [\sum_{k=1}^{N_m} \log p(x_{m,k}|\theta)] \tag{A.10}$$

$$= \frac{d}{d\mu_{q^{(s-1)}}} \mathbb{E}_{q^{(s-1)}} [\log p(x_m|\theta)], \tag{A.11}$$

where the last equality assumes that the data are conditionally independent given the model parameters. □

**Property A.5** (cf. Property 5 of Bui et al. 2018). *Under the assumptions of Prop. A.4, using local averaging with parallel global updates result in identical dynamics for $q(\theta)$, given by the following equation, regardless of the partition of the data employed:*

$$\lambda_q^{(s)} = \lambda_0 + \frac{d}{d\mu_{q^{(s-1)}}} \mathbb{E}_{q^{(s-1)}} [\log p(x|\theta)] = \lambda_0 + \sum_{i=1}^{\sum_j n_j} \frac{d}{d\mu_{q^{(s-1)}}} \mathbb{E}_{q^{(s-1)}} [\log p(x_i|\theta)],$$

*where $x_i$ is the ith data point, and $n_j$ is the number of local samples on client $j$.*

*Proof.* With $M$ clients doing a parallel update, from Property A.4 we have

$$\lambda_q^{(s)} = \lambda_0 + \sum_{j=1}^{M} \lambda_j^{(s)} \tag{A.12}$$

$$= \lambda_0 + \sum_{j=1}^{M} \frac{d}{d\mu_{q^{(s-1)}}} \mathbb{E}_{q^{(s-1)}} [\log p(x_j|\theta)] \tag{A.13}$$

$$= \lambda_0 + \sum_{j=1}^{M} \sum_{k=1}^{n_j} \frac{d}{d\mu_{q^{(s-1)}}} \mathbb{E}_{q^{(s-1)}} [\log p(x_{m,k}|\theta)], \tag{A.14}$$

where Equation A.14 follows due to data being conditionally independent given the model parameters.

On the other hand, with $M = 1$ Property A.4 reads

$$\lambda_q^{(s)} = \lambda_0 + \frac{d}{d\mu_{q^{(s-1)}}} \mathbb{E}_{q^{(s-1)}} [\log p(x|\theta)] \tag{A.15}$$

$$= \lambda_0 + \sum_{i=1}^{n} \frac{d}{d\mu_{q^{(s-1)}}} \mathbb{E}_{q^{(s-1)}} [\log p(x_i|\theta)], \tag{A.16}$$

which matches *Equation A.14*, since $n = \sum_j^M n_j$ for any $M$.

□

**DP with local averaging**

**Theorem A.6.** *Assume the change in the model parameters $\|\lambda_{m_k}^* - \lambda^{(s-1)}\|_2 \le C, k = 1, \ldots, N_m$ for some known constant $C$, where $\lambda_{m_k}^*$ is a proposed solution to Equation 4.1, and $\lambda^{(s-1)}$ is the vector of common initial values. Then releasing $\Delta\hat{\lambda}_m^*$ is $(\varepsilon, \delta)$-DP, with $\delta \in (0,1)$ s.t. $\varepsilon = \mathbb{O}(\delta, q_{sample} = 1, 1, \mathcal{G}_\sigma)$, when*

$$\Delta\hat{\lambda}_m^* = \frac{1}{N_m}\Big[\sum_{k=1}^{N_m}\big(\lambda_{m_k}^* - \lambda^{(s-1)}\big) + \xi\Big], \tag{A.17}$$

*where $\xi \sim \mathcal{N}(0, \sigma^2 \cdot I)$.*

*Proof.* Considering neighbouring datasets as in the DP definition 2, denoted by $m, m'$, only one of the local models is affected by the differing element, w.l.o.g. assume it is $\lambda_{m_1}^*$. For the difference in the sum query between neighbouring datasets we therefore immediately have

$$\|\sum_{k=1}^{N_m}\big(\lambda_{m_k}^* - \lambda^{(s-1)}\big) - \sum_{k=1}^{N_m}\big(\lambda_{m_k'}^* - \lambda^{(s-1)}\big)\|_2 = \|\lambda_{m_1}^* - \lambda^{(s-1)} - \lambda_{m_1'}^* + \lambda^{(s-1)}\|_2 \tag{A.18}$$

$$\le \|\lambda_{m_1}^* - \lambda^{(s-1)}\|_2 + \|\lambda_{m_1'}^* - \lambda^{(s-1)}\|_2 \tag{A.19}$$

$$\le 2C. \tag{A.20}$$

The sum query therefore corresponds to a single call to the Gaussian mechanism with sensitivity $2C$ and noise standard deviation $\sigma$, and since DP guarantees are not affected by post-processing such as taking average, the claim follows. $\square$

**Corollary A.7.** *A composition of $S$ global updates with local averaging using a norm bound $C$ for clipping is $(\varepsilon, \delta)$-DP, with $\delta \in (0,1)$ s.t. $\varepsilon = \mathbb{O}(\delta, q_{sample} = 1, S, \mathcal{G}_\sigma)$.*

*Proof.* Since each global update is DP by Theorem A.6 when we enforce the norm bound by clipping, the result follows immediately by composing over the global updates. $\square$

**Theorem A.8.** *With local averaging, The DP noise standard deviation can be scaled as $\mathcal{O}(\frac{1}{N_m})$, where $N_m$ is the number of local partitions. Therefore, the effect of DP noise will vanish on the local factor level when the local dataset size and the number of local partitions grow.*

*Proof.* Rewriting Equation A.17 as

$$\Delta\hat{\lambda}_m^* = \frac{1}{N_m}\Big(\sum_{k=1}^{N_m}\lambda_{m_k}^* - \lambda^{(s-1)}\Big) + \frac{\xi}{N_m}, \tag{A.21}$$

where $\xi \sim \mathcal{N}(0, \sigma^2 \cdot I)$.

Letting the number of local partitions grow, we immediately have

$$\lim_{N_m \to \infty} \Delta\hat{\lambda}_m^* = \lim_{N_m \to \infty} \Delta\lambda_m^*, \tag{A.22}$$

where $\Delta\lambda_m^*$ is the corresponding non-DP average. $\square$

**Theorem A.9.** *Assume the effective prior $p_{\setminus j}(\eta)$, and the likelihood $p(x_j|\eta), j \in \{1, \ldots, M\}$ are in a conjugate exponential family, where $\eta$ are the natural parameters. Then the number of partitions used in local averaging does not affect the non-DP posterior.*

*Proof.* To avoid notational clutter, we drop the client index $j$ in the rest of this proof, and simply write, e.g., $x$ for the local data $x_j$ and $p(\eta)$ for the effective prior $p_{\setminus j}(\eta)$.

Due to the conjugacy we can write the effective prior and the likelihood as

$$p(x|\eta) = h(x)\exp(\eta^T T(x) - A(\eta)), \tag{A.23}$$

$$p(\eta|\tau_0, n_0) = H(\tau_0, n_0)\exp(\tau_0^T \eta - n_0 A(\eta)), \tag{A.24}$$

where $T$ are the sufficient statistics, $A$ is the log-partition function, $\tau_0, n_0$ are the effective prior parameters, and $h, H$ are some suitable functions determined by the exponential family.

The local posterior given a vector $x$ of $n$ iid observations is in the same exponential family:

$$p(\eta|x, \tau_0, n_0) \propto \exp\left((\tau_0 + \sum_{i=1}^n T(x_i))^T \eta - (n_0 + n)A(\eta)\right),$$

so the changes in parameters when updating from prior to posterior are

$$\tau_0 \to \tau_0 + \sum_{i=1}^n T(x_i) \tag{A.25}$$

$$n_0 \to n_0 + n. \tag{A.26}$$

Now partitioning the data into $N$ shards, each with $n_k = \frac{n}{N}$ samples, together with the tempered (cold) likelihood $p(x|\eta)^N$ results in a posterior

$$p_{shard}(\eta|x_k, \tau_0, n_0) \propto \exp((\tau_0 + \sum_{k_i} NT(x_{k_i}))^T \eta - (n_0 + Nn_k)A(\eta)),$$

so the updates for shard $k$ are

$$\tau_0 \to \tau_0 + \sum_{k_i} NT(x_{k_i}) \tag{A.27}$$

$$n_0 \to n_0 + Nn_k. \tag{A.28}$$

Averaging over the posterior parameters corresponding to the $N$ local data shards we have updates

$$\tau_0 \to \frac{1}{N} \sum_{k=1}^N \left(\tau_0 + \sum_{k_i} NT(x_{k_i})\right) \tag{A.29}$$

$$= \tau_0 + \sum_{k=1}^N \sum_{k_i} T(x_{k_i}) \tag{A.30}$$

$$= \tau_0 + \sum_{i=1}^n T(x_i), \text{ and} \tag{A.31}$$

$$n_0 \to \frac{1}{N} \sum_{k=1}^N \left(n_0 + Nn_k\right) \tag{A.32}$$

$$= n_0 + n, \tag{A.33}$$

which match the expressions for the regular local posterior using full local data given in Equation A.25 and Equation A.26. □

Figure 5 shows the effects of changing the number of local data shards using a logistic regression model with and without DP. As discussed in Section 4.2.1, when the assumptions of Theorem A.9 are not satisfied, we would expect that increasing the number of local data partitions can lead to slower convergence due to increased estimator variance. This can be seen in Figure 5 a). In contrast, under privacy constraints increasing the number of local partitions can lead to improved performance, since adding local partitioning

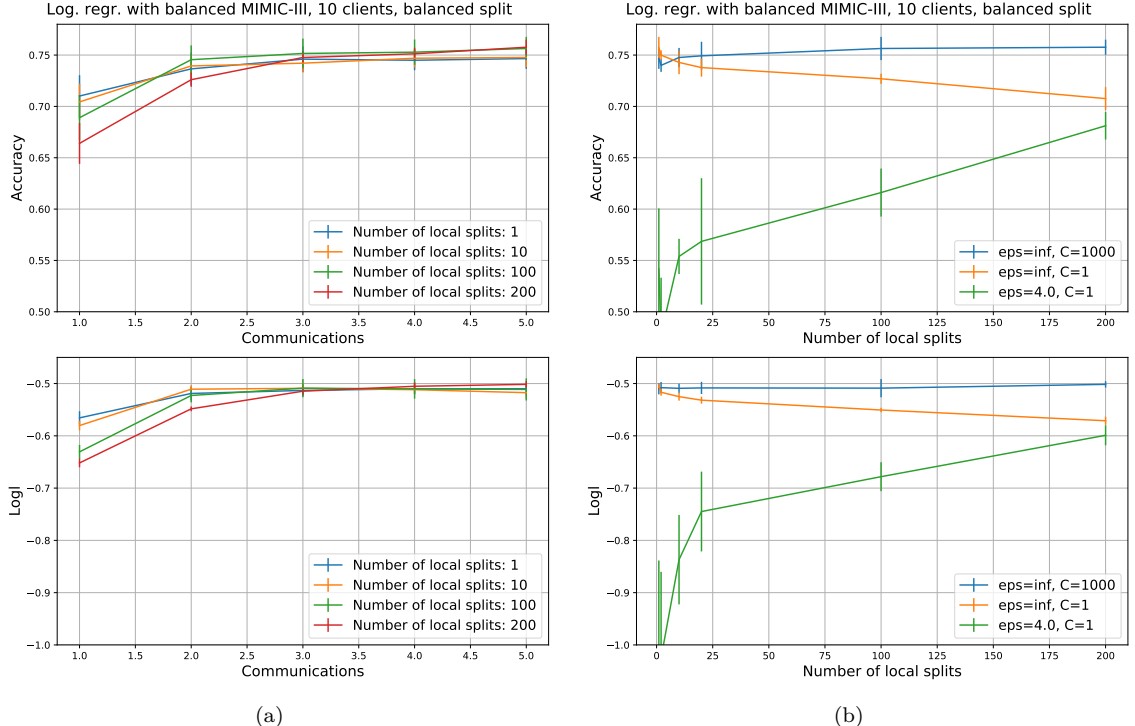

(a)                                                             (b)

Figure 5: Logistic regression, balanced MIMIC-III data with 10 clients: mean over 5 seeds with SEM, balanced split. a) Without DP, increasing the number of local partitions can lead to slower convergence, b) non-DP with clipping norm $C$, and $(4, 10^{-5})$-DP: wihout privacy increasing the number of local partitions does not help (non-DP with clipping $C = 1000$), or even hurts performance (non-DP with clipping $C = 1$), while with DP, increasing the number of local partitions mitigates the effect of DP noise.

can mitigate the DP noise effect, as seem in Figure 5 b). Note that increasing the number of local partitions can also increase the bias due to clipping, especially with tight clipping bound. In this experiment, we use same fixed hyperparameters in all runs: number of global updates or communication rounds $= 5$, number of local steps $= 50$, learning rate $= 10^{-2}$, damping $= .4$.

**Theorem A.10.** *Using local averaging with $M$ clients and a shared number of local partitions $N_j = N \; \forall j$ assume the clients have access to a trusted aggregator. Then for any given privacy parameters $\varepsilon, \delta$, the noise standard deviation added by a single client can be scaled as $\mathcal{O}(\frac{1}{\sqrt{M}})$ while guaranteeing the same privacy level.*

*Proof.* Let $\eta \sim \mathcal{N}(0, \sigma^2 \cdot I)$, and denote by $\sigma_0$ the noise standard deviation that locally guarantees the required DP level for every client with some known norm bound $C$ (possibly due to clipping), and assign equal noise shares over clients. The message for a synchronous global update $s$ is

$$\prod_{j=1}^{M} \Delta t_m^{(s)} = \sum_{j=1}^{M} \left( \frac{1}{N} \left[ \sum_{k=1}^{N} (\lambda_{j_k}^* - \lambda^{(s-1)}) + \eta \right] \right) \tag{A.34}$$

$$= \frac{1}{N} \left( \sum_{j=1}^{M} \sum_{k=1}^{N} (\lambda_{j_k}^* - \lambda^{(s-1)}) + \sum_{j=1}^{M} \eta \right). \tag{A.35}$$

To match the target local noise standard deviation with the aggregated noise standard deviation we need

$$\sum_{j=1}^{M} \sigma^2 \geq \sigma_0^2 \tag{A.36}$$

$$\Leftrightarrow \sigma \geq \frac{\sigma_0}{\sqrt{M}}. \tag{A.37}$$

Setting $\sigma$ to match the lower bound, we see that the total noise magnitude on the global approximation level in Equation A.35 does not change with $M$. $\qquad\square$

Looking at Theorem A.10, when the global noise level is constant, adding a client to the protocol will reduce the relative effect of the noise in Equation A.35 if it increases the non-noise part in the sum. On the other hand, on global convergence we would have

$$\sum_{j=1}^{M} \left( \frac{1}{N} \sum_{k=1}^{N} \lambda_{j_k}^* - \lambda^{(s-1)} \right) = 0,$$

so an update near a global optimum will be mostly noise.

When applying Theorem A.10, DP is guaranteed jointly on the global model level, while the local approximations have less noise than required for the stated privacy level (although they might still have some valid DP guarantees). In contrast, when each client guarantees DP independently via Theorem A.6, the global model will be $(\epsilon_{max}, \delta_{max})$-DP w.r.t. any single training data sample by parallel composition with $\epsilon_{max} = \max\{\epsilon_1, \ldots, \epsilon_M\}, \delta_{max} = \max\{\delta_1, \ldots, \delta_M\}$, where $\epsilon_m, \delta_m$ are the parameters used by client $m$. In all the experiments in this paper, we use a common $(\epsilon, \delta)$ budget shared by all the clients.

**DP with virtual PVI clients**

**Theorem A.11.** *Assume the change in the model parameters $\|\lambda_{m_k}^* - \lambda^{(s-1)}\|_2 \leq C, k = 1, \ldots, N_m$ for some known constant $C$, where $\lambda_{m_k}^*$ is a proposed solution to Equation 4.8, and $\lambda^{(s-1)}$ is the vector of common initial values. Then releasing $\Delta\tilde{\lambda}_m^*$ is $(\varepsilon, \delta)$-DP, with $\delta \in (0, 1)$ s.t. $\varepsilon = \mathbb{O}(\delta, q_{sample} = 1, 1, \mathcal{G}_\sigma)$, when*

$$\Delta\tilde{\lambda}_m^* = \sum_{k=1}^{N_m} \left( \lambda_{m_k}^* - \lambda^{(s-1)} \right) + \eta, \tag{A.38}$$

*where $\eta \sim \mathcal{N}(0, \sigma^2 \cdot I)$.*

*Proof.* Almost the same as the proof of Theorem A.6. $\qquad\square$

**Corollary A.12.** *A composition of $S$ global updates with virtual PVI clients using a norm bound $C$ for clipping is $(\varepsilon, \delta)$-DP, with $\delta \in (0, 1)$ s.t. $\varepsilon = \mathbb{O}(\delta, q_{sample} = 1, S, \mathcal{G}_\sigma)$.*

*Proof.* Since each global update is DP by Theorem A.11 when we enforce the norm bound by clipping, the result follows immediately by composing over the global updates. $\qquad\square$

**Theorem A.13.** *Assume there are $M$ real clients adding virtual clients, and access to a trusted aggregator. Then for any given privacy parameters $\varepsilon, \delta$, the noise standard deviation added by a single client can be scaled as $\mathcal{O}(\frac{1}{\sqrt{M}})$ while guaranteeing the same privacy level.*

*Proof.* Similar to the proof for Theorem A.10 with obvious modifications. $\qquad\square$

As with local averaging, Theorem A.13 gives joint DP guarantees on the global model level. In contrast, when each client guarantees DP independently with Theorem A.11, the global model will be $(\epsilon_{max}, \delta_{max})$-DP w.r.t. any single training data sample by parallel composition with $\epsilon_{max} = \max\{\epsilon_1, \ldots, \epsilon_M\}, \delta_{max} = \max\{\delta_1, \ldots, \delta_M\}$, where $\epsilon_m, \delta_m$ are the parameters used by client $m$. And again, in all the experiments we use a common $(\epsilon, \delta)$ budget shared by all the clients.

## B   Appendix: experimental details

This appendix contains details of the experimental settings omitted from Section 5.

With Adult data, we first combine the training and test sets, and then randomly split the whole data with 80% for training and 20% for validation. With MIMIC-III data, we first preprocessing the data for the in-hospital mortality prediction task as detailed by Harutyunyan et al. (2019).[7]. Since the preprocessed data is very unbalanced and leaves little room for showing the differences between the methods (a constant prediction can reach close to 90% accuracy while a non-DP prediction can do some percentage points better), we first re-balance the data by keeping only as many majority label samples as there are in the minority class. This leaves 5594 samples, which are then randomly split into training and validation sets, giving a total of 4475 samples of training data to be divided over all the clients.

We divide the data between $M$ clients using the following scheme[8]: half of the clients are small and the other half large, with data sizes given by

$$n_{small} = \big\lfloor \frac{n}{M}(1 - \rho) \big\rfloor, \quad n_{large} = \big\lfloor \frac{n}{M}(1 + \rho) \big\rfloor,$$

with $\rho \in [0, 1]$. $\rho = 0$ gives equal data sizes for everyone while $\rho = 1$ means that the small clients have no data. For creating unbalanced data distributions, denote the fraction of majority class samples by $\lambda$. Then the target fraction of majority class samples for the small clients is parameterized by $\kappa$:

$$\lambda_{small}^{target} = \lambda + (1 - \lambda) \cdot \kappa,$$

where having $\kappa = 1$ means small clients only have majority class labels, and $\kappa = -\frac{\lambda}{1-\lambda}$ implies small clients have only minority class labels. For large clients the labels are divided randomly.

We use the following splits in the experiments:

|  | $\rho$ | $\kappa$ | $n_{small}$ | $\lambda_{small}$ | $\lambda_{large}$ |
|---|---|---|---|---|---|
| balanced | 0 | 0 | 2442 | .76 | $\simeq .76$ |
| unbalanced 1 | .75 | .95 | 610 | .99 | $\simeq .73$ |
| unbalanced 2 | .7 | -3 | 732 | .03 | $\simeq .89$ |

Table 2:   Adult data, 10 clients data split.

|  | $\rho$ | $\kappa$ | $n_{small}$ | $\lambda_{small}$ | $\lambda_{large}$ |
|---|---|---|---|---|---|
| balanced | 0 | 0 | 122 | .75 | .7-.8 |
| unbalanced 1 | .75 | .95 | 30 | .97 | .67-.78 |
| unbalanced 2 | .7 | -3 | 36 | .03 | .85-.93 |

Table 3:   Adult data, 200 clients data split.

|  | $\rho$ | $\kappa$ | $n_{small}$ | $\lambda_{small}$ | $\lambda_{large}$ |
|---|---|---|---|---|---|
| balanced | 0 | 0 | 447 | .5 | $\simeq .5$ |
| unbalanced 1 | .75 | .95 | 111 | .97 | .41-.44 |
| unbalanced 2 | .7 | -.5 | 134 | .25 | .51-.58 |

Table 4:   Balanced MIMIC-III data, 10 clients data split.

We use Adam (Kingma & Ba, 2014) to optimise all objective functions. In general, depending e.g. on the update schedule, even the non-DP PVI can diverge (see Ashman et al. 2022). We found that DP-PVI using

---

[7]The code for preprocessing is available from `https://github.com/YerevaNN/mimic3-benchmarks`

[8]The data splitting scheme was originally introduced by Sharma et al. (2019) in a workshop paper that combines DP with PVI. The current paper is otherwise completely novel and not based on the earlier workshop version.

any of our approaches is more prone to diverge than non-DP PVI, while DP optimisation is more stable than local averaging or virtual PVI clients. To improve model stability, we use some damping in all the experiments. When damping with a factor $\rho \in (0, 1]$, at global update $s$ the model parameters $\lambda^{(s)}$ are set to

$$(1 - \rho) \cdot \lambda^{(s-1)} + \rho \cdot \lambda^{(s)}.$$

We use grid search to optimise all hyperparameters in terms of predictive accuracy and model log-likelihood using 1 random seed, and then run 5 independent random seeds using the best hyperparameters from the 1 seed runs. The reported results with 5 random seeds are the best results in terms of log-likelihood for each model. With BNNs using local averaging or virtual PVI clients some seeds diverged when using the hyperparameters optimised using a single seed. These seeds were rerun with the same hyperparameter settings to produce 5 full runs. This might give the methods some extra advantage in the comparison, but since they still do not work too well, we can surmise that the methods are not well suited for the task.

To approximate the posterior predictive distributions we use a Monte Carlo estimate with 100 samples:

$$p(y_* | x_*, y, x) \simeq \frac{1}{100} \sum_{i_{MC}=1}^{100} p(y_* | x_*, \theta_{i_{MC}}), \quad \theta_{i_{MC}} \sim q(\theta).$$

