# OpenReview forum: "Differentially private partitioned variational inference"
_TMLR — Accepted by TMLR_

### Review · Reviewer_MNCv · 2022-10-27

**Summary Of Contributions:**

This paper studies the problem of privacy preserving federated learning for probabilistic models. Essentially, the paper considers a setting where several users hold disjoint datasets and aim to collaboratively learn a probabilistic model while enforcing central differential privacy on their joint dataset. In short, the paper proposes three different solutions to solve this problem, all based on adapting a model called Partitioned Variational Inference (PVI) with a Gaussian mechanism (well known from the differential privacy community). The three methods are claimed to provide differential privacy and are compared empirically on two simple datasets with a logistic regression model and a Bayesian neural network.

Before starting the review, I just want to clearly state that I am not an expert in variational inference. I consider myself more knowledgeable in privacy-preserving distributed/federated learning, especially when considering differential privacy.

**Audience:**

No

**Broader Impact Concerns:**

I have no broader impact concern

**Claims And Evidence:**

No

**Requested Changes:**

In view of the above remarks, I have two requests that are essential for me to recommend acceptance.

1. Rework pages 2-9 to incorporate the above comments on formalism and improve the technical presentation of solutions and claims.

2. Clearly present the main findings of the paper and how they differ from the existing literature. In doing so, I would recommend moving the related work section closer to the introduction, and comparing the contributions with the literature directly at the beginning of the paper.

**Strengths And Weaknesses:**

As I see it, the main weakness of the paper is the technical soundness. I think the paper needs to be considerably reworked regarding the presentation of both the proposed solutions and the technical claims. As is, it is very difficult for me to assess the technical quality of the claims. I provide a list of some of my issues below to justify this concern.

- Throughout the paper, the elements of the problem ($\theta$,$x$, $x_i$, $p$,$q$, $\lambda, t_i$, etc.) are introduced in words but without describing their mathematical nature (scalars, vectors in $\mathbb{R}^d$, distributions with support in a given vector space, mappings, etc.). To my point of view, the absence of a formal and clear problem statement in the beginning of the paper obfuscates the discourse as it introduces potential ambiguities that hinder the assessment of the technical soundness of the work.
- Note also that the minimization/maximization problems suffer from the same ambiguity. One the one hand, the optimization problems are defined by taking $q \in \mathcal{Q}$, where $\mathcal{Q}$ is a class of distributions. On the other hand, the selected argmax in (2.1) and throughout the paper is a variational parameter that I guess is a real-valued vector. This kind of typo is obviously not critical in general, but at this stage I think their accumulation also obfuscates the readability. I list below some more inconsistencies that I think should be fixed:
  - $q$ should be defined in a consistent manner, either $q(. \mid \lambda)$ or $q(.)$.
  - The Kullback-Leibler notation should be consistent.
  - $\Delta t_m^{(i)}$ and $q_m^*$ should be defined before being used in Algo 1 line 4.
  - $q^*$ should be defined before being used in Algo 3 line 5.
  - $\eta$ is used to describe different elements (natural parameters in Theorem 7 and noise in Theorems 4 and 9).
- The technical content, be it definitions or theorems does not seem to rest on solid ground. To me, this comes from the fact that the results and definitions are formulated in words and refer to concepts that have not been introduced in the paper. I list some examples below.
  - In Definition 2, the accounting oracle is defined by referring to the ‘subsampled mechanism’ and a ‘subsampling ratio’ that have not been introduced in the paper. These concepts are used throughout the entire technical part of the paper without being ever defined. More importantly, the proposed solutions all build on the Gaussian mechanism (and DP-SGD for Theorem 3) that is not formalized either. While all these concepts are standard in the differential privacy literature, restating them with the paper’s formalism would raise a number of ambiguities.
  - The theorems are not self-contained and assume some a priori knowledge from the reader. For examples, Theorem 4 and Theorem 9 (that are supposed to be treating different solutions) are stated in the exact same way, making them identical unless assuming a priori knowledge from the reader. I strongly believe that each solution should be formally presented (e.g., in a small algorithm or in a sequence of equations) and that each theorem should reference the solution it applies to.
  - In paragraph ‘DP with local averaging’ and in Theorem 7, the paper mentions ‘exponential family factors/parameters’ that are not formally described. I think it should be clarified especially since exponential families have very versatile forms; hence it is primary to clearly define which form is being used.

Another important concern I have is on whether the paper advances the state of knowledge in terms of privacy-preserving distributed models. As I understand it, the main takeaway of the paper consists in saying that (i) applying differential privacy on a local optimizer is primarily interesting because it benefits from the size of the local datasets, and (ii) applying differential privacy when merging the local updates, can also benefit from having access to a trusted aggregator (e.g., a shuffler). As far as my understanding of the existing literature goes, I believe that these findings do not provide additional intuition to the community. Specifically, the importance of the size of the dataset (and thus of the mini-batches of data) when applying DP-SGD is already well documented in (Song et al., 2013; Bassily et al., 2014; Abadi et al., 2016)(*). Similarly, the importance of a shuffler in improving the performances of a differentially private federated learning scheme has been documented in e.g., (Cheu et al., 2019;  Erlingsson et al., 2019). To be clear, I am not saying that the paper does not provide any new intuition. My point is rather that, in the absence of a clear discussion with existing wisdom from the literature on differential privacy in (federated/distributed) machine learning, it is challenging for me to grasp the nature of this contribution.

(*) These works are referenced in the paper

Cheu, A., Smith, A., Ullman, J., Zeber, D., Zhilyaev, M. (2019). Distributed Differential Privacy via Shuffling. EUROCRYPT 2019.
Erlingsson, A., Feldman V., Mironov I., Raghunathan A., Talwar K., and Thakurta, A. (2019). Amplification by Shuffling: From Local to Central Differential Privacy via Anonymity. SODA 2019.

---

> ### Author Response · Authors · 2023-01-17
> **Reply to reviewer MNCv 1/2**
>
> We thank the reviewer for their feedback and effort. The following are replies to specific points in the review.
>
> 1.
>
> > Throughout the paper, the elements of the problem are introduced in words but without describing their mathematical nature
>
> We have added several explanations and more explicit definitions throughout the paper, as well as a note about using real-valued data in the experiments to make it more readable. We generally use standard notation; e.g., for writing general probability distributions we simply write p (and overload the notation in a standard way to identify prior, likelihood and the posterior by their arguments) without defining the distribution explicitly, if all we want to say is that we assume some distribution. Similarly, we generally do not require, e.g., real-valued data for developing the theory; any data over which we can define suitable probability distributions will work.
>
> 2.
>
> > the absence of a formal and clear problem statement in the beginning of the paper obfuscates the discourse as it introduces potential ambiguities that hinder the assessment of the technical soundness of the work
>
> We introduce the general problem in words in the Introduction, and more formally in reviewing the (non-DP) PVI method and DP in the Background section. Hopefully, this is now clearer after some added explanations and several fixed typos.
>
> 3.
>
> > Note also that the minimization/maximization problems suffer from the same ambiguity.
>
> We have fixed the typos and confusing changes in the notation so hopefully this is now more understandable. The main point in the VI approximation is that we pick some tractable family of distributions Q, and try to find the best approximation within this family. Within the family Q, the actual approximating distribution q(.|lambda) is a distribution indexed by the parameter vector lambda. When doing the optimisation, we optimise this parameter vector lambda, so we basically find better approximating distribution q from the family Q.
>
> 4.
>
> > The technical content, be it definitions or theorems does not seem to rest on solid ground. To me, this comes from the fact that the results and definitions are formulated in words and refer to concepts that have not been introduced in the paper.
>
> We had skipped repeating some standard definitions (such as the Gaussian mechanism for DP) to keep the paper more concise. We have now added several such definitions in an effort to make the paper easier to read (Gaussian mechanism, subsampling, exponential family in Sections 3.1 and 3.2).
>
> 5.
>
> > ‘subsampled mechanism’ and a ‘subsampling ratio’ ... have not been introduced in the paper. ... [T]he proposed solutions all build on the Gaussian mechanism ... that is not formalized either.
>
> We have added more formal definitions for these (Section 3.2).
>
> 6.
>
> > The theorems are not self-contained and assume some a priori knowledge from the reader.
>
> This is fixed (we added references to Theorems 7 and 12 pointing to the specific Algorithms in question).

---

> > ### Author Response · Authors · 2023-01-17
> > **Reply to reviewer MNCv 2/2**
> >
> > 7.
> >
> > > In paragraph ‘DP with local averaging’ and in Theorem 7, the paper mentions ‘exponential family factors/parameters’ that are not formally described.
> >
> > We added a definition for the exponential family (Section 3.1).
> >
> > 8.
> >
> > > As I understand it, the main takeaway of the paper consists in saying that (i) applying differential privacy on a local optimizer is primarily interesting because it benefits from the size of the local datasets, and (ii) applying differential privacy when merging the local updates, can also benefit from having access to a trusted aggregator (e.g., a shuffler).
> >
> > We consider the main takeaway from the paper to be that we provide concrete implementations as well as some theoretical understanding for running a Bayesian federated learning method with differential privacy, more specifically, DP-PVI. This allows us to approximate any posterior distributions in the federated learning setting via DP-PVI, and hopefully to also have some reasonable expectations on whether or not this is a worthwhile idea with a given model and data set. As for the main differences between our proposed implementations, we agree with your takeaway.
> >
> > 9.
> > > As far as my understanding of the existing literature goes, I believe that these findings do not provide additional intuition to the community.
> >
> > As far as we are aware, there is no prior work on DP-PVI, or on DP VI focusing on the federated learning setting. From the references mentioned in the review, Song et al., 2013; Bassily et al., 2014; Abadi et al., 2016 all focus on formulating or running DP-SGD on a single centralised data set to minimise a given loss function, while Cheu et al., 2019; Erlingsson et al., 2019 formalise the shuffle DP privacy model, and show privacy amplification results due to the shuffler. None of these make any mention of Bayesian learning or approximating posteriors via VI.
> > While our results do agree, e.g., with the general intuition that more data tends to help,
> > we believe that there is significant value for the community in actually formulating and empirically testing methods in this setting.
> >
> > 10.
> >
> > > Rework pages 2-9 to incorporate the above comments on formalism and improve the technical presentation of solutions and claims.
> >
> > We have clarified the text (e.g., we fixed notations for consistency, such as q() vs q(|lambda), added some explanatory remarks, such as about real-valued data used in the experiments, and fixed overlapping notations for noise and natural parameters), added several previously omitted definitions (Gaussian mechanism, subsampling and exponential families), as well as fixed plenty of typos. Hopefully this makes the paper much clearer.
> >
> > 11.
> >
> > > Clearly present the main findings of the paper and how they differ from the existing literature. In doing so, I would recommend moving the related work section closer to the introduction, and comparing the contributions with the literature directly at the beginning of the paper.
> >
> > As commented above, we are not aware of any closely-related existing work that would be sensible as a comparison. We have moved Related work to the beginning of the paper.

---

### Review · Reviewer_gi6N · 2022-12-06

**Summary Of Contributions:**

The paper considers adding differential privacy guarantees to the problem of Partitioned Variational Inference in the federated learning setting. They do so by adopting a very recent paper by Ashman et al. that proposed a framework in FL setting.

The contributions by the authors include when DP guarantees are applied: at the client or at the server and how it is added at the client (either through the update or through computing local models).

The paper also supplements its results with experiments on UCI adult dataset to compare the methods in data that may not have the original assumptions.

**Audience:**

No

**Claims And Evidence:**

Yes

**Requested Changes:**


- introduction of the paper does not motivate the problem and the result well. Which setting is the result important? What was challenging about it? How it is different from Ashman et al. 2022?
- what does it mean for the work to be communication efficient? It seems it is an empirical results and not theoretical. this should be clearly stated.
- the problem setting is undefined. What is the threat model (e.g., saying clients are honest is not sufficient, as it seems DP guarantees would change depending on which proposed method is used)? How can there be no trusted aggregator? what is meant by secure primitive?
- Which dataset neighbouring definition is relevant for which section? is it user or sample DP? Seems it would change depending on the method being used?
- Notation needs to be clearly stated, how many clients, what data they have, assumptions on the data. client m vs client j. How are sample, individual, party and client related across the paper? These are crucial to communicate which DP property is guaranteed (local vs global)
- Table 1 and statements before Section 3.2. are quite vague. For example, what does it mean to benefit from trusted Aggregator? How did one arrive to this table, what should reader take from it?
- theorem statements are not self-inclusive. Theorem 11 does not say what level of DP guarantee this would provide.


**Strengths And Weaknesses:**

Strength:
- Federated learning with DP guarantees is an important problem
- Experimental results

Weaknesses:
- the writing could be significantly improved (starting from the introduction to motivate the problem)
- formal definition of neighbouring is missing
- notation keeps changing throughout the paper
- threat model and guarantees are not clear to compare the three methods

---

> ### Author Response · Authors · 2023-01-17
> **Reply to the review for gi6N 1/2**
>
> We thank the reviewer for their feedback and effort. The following are replies to specific points in the review.
>
> 1.
>
> > the writing could be significantly improved
>
> > notation keeps changing throughout the paper
>
> We have clarified the notation, added some more explanations throughout the article, as well as fixed plenty of typos, in an effort to make the article clearer and easier to read (e.g., we added definitions for Gaussian mechanism, subsampling and exponential families, fixed notations for consistency, such as q() vs q(|lambda), added some explanatory remarks, such as about real-valued data used in the experiments in Section 3.1, and fixed overlapping notations for noise and natural parameters).
>
> 2.
>
> > How it is different from Ashman et al. 2022?
>
> Ashman et al. 2022 introduce non-private PVI. Our work builds on their PVI framework, but we focus on guaranteeing DP in learning: it is generally not obvious how to do this, and not obvious what properties may hold from the non-private algorithm. Our paper explores this and makes this vital contribution.
>
> 3.
>
> > formal definition of neighbouring is missing
>
> We added explicit comment in defining DP that we use the bounded neighbourhood relation.
>
> 4.
>
> > what does it mean for the work to be communication efficient? It seems it is an empirical results and not theoretical. this should be clearly stated.
>
> We measure communication-efficiency as number of client-server communication rounds; we added a clarification on the experimental evidence to the contribution statement.
>
> 5.
>
> > threat model and guarantees are not clear to compare the three methods
>
> > the problem setting is undefined. What is the threat model (e.g., saying clients are honest is not sufficient, as it seems DP guarantees would change depending on which proposed method is used)?
> ... Which dataset neighbouring definition is relevant for which section? is it user or sample DP? Seems it would change depending on the method being used?
>
> We are interested in providing privacy for learning the global model w.r.t. training data. We want to protect individuals present in the training data, assuming each individual has a single sample in the full combined training data, i.e., we use sample-level neighbourhood granularity (Section 3.2, start of Section 4). This does not change between the methods. When there is no secure primitive available, each client guarantees DP independently, which then implies DP for the global model. With access to a trusted aggregator, the global model DP guarantees are joint guarantees, since they depend on the noise contributions from all of the clients (see the comments after the proofs of Theorems A.1, A.10 and A.13 in the Appendix, the comments about Table 1 in Section 4, and the comments about Theorems 6, 11, and 14 in the technical summary in Section 4.3).

---

> > ### Author Response · Authors · 2023-01-17
> > **Reply to the review for gi6N 2/2**
> >
> > 6.
> >
> > > what is meant by secure primitive? How can there be no trusted aggregator?
> >
> > Secure primitives or cryptographic primitives are a common term for basic cryptographic algorithms, which allow for computing some function securely. Examples relevant to the current work include, e.g., secure sum or secure shuffling. We consider learning both without assuming any secure primitives, and assuming a secure sum primitive (simulated by the clients submitting values to a trusted aggregator). Secure primitives typically require significantly more computational and/or communication resources (see e.g. Lindell & Pinkas 2009: Secure multiparty computation for privacy-preserving data mining for a nice general discussion), so we also want to test how the methods perform without any secure primitives.
> >
> > 7.
> >
> > > Notation needs to be clearly stated, how many clients, what data they have, assumptions on the data. client m vs client j. How are sample, individual, party and client related across the paper? These are crucial to communicate which DP property is guaranteed (local vs global)
> >
> > The parties involved in the federated learning protocol are the clients and the server. We assume M clients, with client j having n_j > 0 samples. The data samples are about some individuals (1 per individual), that we want to protect with DP. Indices j and m are used as (changing) indices for the clients that depend on the context. We do not generally require any specific structure for the data, since we mostly work with probability distributions in developing the necessary theory. However, in all the experiments the data is in R^d with some d. We added a note on this to the article at the start of Section 3.1.
> >
> > 8.
> >
> > > Table 1 and statements before Section 3.2. are quite vague. For example, what does it mean to benefit from trusted Aggregator? How did one arrive to this table, what should reader take from it?
> >
> > Table 1 highlights the properties that are discussed more formally in the following sections. By benefiting from a trusted aggregator we mean that we can formulate joint DP guarantees, where the noise level for each individual client is less than if each client guarantees DP independently (see reference to Table 1 in Section 4).
> >
> > 9.
> >
> > > theorem statements are not self-inclusive.
> >
> > > Theorem 11 does not say what level of DP guarantee this would provide.
> >
> > This is fixed (with the updated numbering, we added references to Theorems 7 and 12 pointing to the specific Algorithms in question, and by making the privacy parameters explicit in Thms 11 and 14).

---

### Review · Reviewer_HTxx · 2023-01-09

**Summary Of Contributions:**

The paper studies communication-efficient variational inference (VI) in a client-distributed setting under client privacy constraints. In this setting, a central model is updated based on the learning of several clients who do not share their data with each other, only privacy-preserving statistics thereof. The goal is to learn an accurate model in as few communication rounds as possible while maintaining privacy. The work builds on partitioned VI (PVI, Ashman et al., 2022) and differential privacy (DP) to produce three different methods toward this goal: DP optimisation, Local averaging, and Virtual PVI Clients. Each algorithm is guaranteed to satisfy DP by applying the so-called Gaussian mechanism to different stages of the learning procedure. Theoretical analyses describe how the magnitude of perturbations necessary to guarantee DP scales with the number of clients and other problem parameters. In an empirical evaluation comparing the accuracy of fitted models, the new methods compare favorably to a sound, yet pessimistic, baseline, but it is less clear which among the three contributions is preferable in which setting.

Claimed contributions:
- Three new algorithms for communication-efficient DP VI
- The new algorithms are orders of magnitude more efficient than Global DP VI, without secure primitives.
- A comparison of the proposed algorithms in terms of their dependence on local data, model class and access to a trusted aggregator

**Audience:**

Yes

**Claims And Evidence:**

Yes

**Requested Changes:**

- Empirical evaluation of the algorithms on data that has inherent client structure.

- Minor issue: The comments surrounding eq (2.2) are imprecise. "However, since the minimisation in Equation 2.1 is still generally intractable, the actual optimisation is typically done by maximising the so-called evidence lower bound. ... It can be shown that the optimal solution λ∗V I also solves the original minimization problem in Equation 2.1.". If solving 2.2 solves 2.1, either 2.1 is not intractable or 2.2 must be intractable as well, in which case tractability can not be the reason to use it.

**Strengths And Weaknesses:**

## Strengths:

- The paper introduces the reader to necessary background on (partitioned) variational inference and differential privacy.
- Contributions are clearly and accurately stated and followed up on.

## Weaknesses:

- This paper combines a number of established ideas into one setting---variational Bayes + differential privacy + partitioned/distributed/federated learning. As a result, the algorithmic contributions are rather small, making minor changes to the existing PVI algorithm or using the DP-SGD optimiser as-is. Similarly, the theoretical properties stated about the algorithms follow largely from existing analysis, such as in the case of Virtual PVI for which "the regular PVI properties ... hold immediately" or the multiple proofs in the appendix with strong reliance on previous works. The DP guarantees also follow standard arguments using clipping and Gaussian noise. Comparatively, Theorems 7 and 8 seem more specific to the combined setting, but are not discussed much in the main paper.

- The empirical evaluation only studies artificial splits of existing batch datasets. This is unfortunate, since one of the paper's main points is that there are problem settings in which we can't perform normal DP VI, but should work in a partitioner or distributed setting. It should be possible to use data with an inherent client notion, such as the eICU data set from the same source as MIMIC (a single-site data set). Moreover, the synthetic splits are all uniform which is highly unrealistic for real settings where different clients are much more likely to have different distributions over their samples (as in the application proposed by the authors)

- There are parameters in the proposed algorithms which are not explored in the empirical study, such as the number of virtual clients/partitions or the privacy parameters. Is the former uninteresting in light of e.g., Theorem 7? What about the effects of a high/low \epsilon? Could the naive parameter perturbation method not be used as a baseline in the experiments?

---

> ### Author Response · Authors · 2023-01-17
> **Reply to the review**
>
> We thank the reviewer for their feedback and effort. The following are replies to specific points in the review.
>
> 1.
> > the algorithmic contributions are rather small, making minor changes to the existing PVI algorithm or using the DP-SGD optimiser as-is ... Similarly, the theoretical properties stated about the algorithms follow largely from existing analysis.
>
> Our main aim is to provide feasible methods for approximating posteriors via privatised PVI, not to introduce algorithmic or theoretical complexity for the sake of novelty. From this perspective, the fact that some of the existing theory can be leveraged directly is a positive aspect, not a weakness. We agree with the reviewer that plugging in DP-SGD instead of regular SGD is not a major algorithmic innovation. Nevertheless, since we know (and again establish in this paper) that DP-SGD tends to be a well-performing general method for DP learning, we consider it essential to include it in a discussion about possible implementation methods, as well as to compare the other approaches to DP-SGD.
>
>
> 2.
> > It should be possible to use data with an inherent client notion.
>
> > Empirical evaluation of the algorithms on data that has inherent client structure.
>
> It is certainly true that we could also use data with inherent client splits. However, in order to empirically test how the number of samples per client and the number of clients affects the proposed algorithms, we think it is clearer to use fixed data sets and simply divide them into differing splits, instead of using completely different data sets with differing inherent splits. Splitting originally centralised data sets for testing is also a common approach in the existing literature (see e.g. Ashman et al. 2022: PVI, Asi et al. 2019: Element level DP, Truex et al. 2019: A hybrid approach to privacy-preserving federated learning, Wei et al. 2019: Federated learning with DP).
>
>
> 3.
> > the synthetic splits are all uniform which is highly unrealistic
>
> We use balanced as well as two unbalanced data splits in the empirical tests. The unbalanced splits are highly non-uniform between the classes of clients. The details for the exact splits for each data set can be found in the Appendix. This is also consistent with the most closely-related existing work (Ashman et al. 2022).
>
>
> 4.
> > There are parameters in the proposed algorithms which are not explored in the empirical study, such as the number of virtual clients/partitions or the privacy parameters. Is the former uninteresting in light of e.g., Theorem 7?
>
> Since the models in the Experiments Section do not satisfy the assumptions of Thm 7 (in the first submission pdf), the number of partitions does have an effect on the results. In the experiments, we treat the number of local data partitions simply as another hyperparameter.
> We have added some more discussion together with some experiments on the effect of this parameter in the Appendix (see comments after Theorem A.9 and Figure 5).
>
>
> 4.
> > What about the effects of a high/low \epsilon?
>
> The privacy parameters directly affect the noise level for the Gaussian mechanism. Based on our tests, the general effect is as would be expected, i.e., when epsilon grows the performance of all our methods improves. While for simple-enough models they seem to approach approximately the same non-DP baseline level, this is not generally guaranteed for arbitrary models. On the other hand, with decreasing epsilon, all our methods will break down eventually when the noise level is high-enough. We have chosen the reported epsilons such that they will guarantee some meaningful level of privacy, as well as to highlight the strong and the weak points in our proposed approaches.
>
>
> 5.
> > Could the naive parameter perturbation method not be used as a baseline in the experiments?
>
> The local averaging and the virtual PVI methods are both generalisations of the naive parameter perturbation in the sense that they are all equal, if we use only a single local data split. With the privacy parameters currently used in the experiments, using a single local split adds so much noise that the naive parameter perturbation often diverges, making it very difficult to use as a general baseline (for comparison, see the results in Figure 5b in the Appendix with looser privacy level epsilon=4).
>
> 6.
> > The comments surrounding eq (2.2) are imprecise
>
> This is fixed (removed the claim about intractability, the equation in question is now 3.3 in the fixed pdf): "optimisation problem ... is usually still not easy-enough for solving directly, the actual optimisation is typically done by maximising the so-called evidence lower bound ..."

---

### Author Response · Authors · 2023-01-17
**Updated paper draft and comments**

We thank all the reviewers for their efforts and detailed feedback. We have incorporated the points raised, and believe that the resulting paper is much improved and more enjoyable to read as a result. We reply to specific points in the reviews separately for each reviewer.
We welcome any additional feedback / discussion.

---

### Author Response · Authors · 2023-02-22
**Inquiring about the decision**

Dear AE and reviewers,

it would be nice to have either a decision on the paper or further comments for discussion, in case the reviewers still have doubts. We believe we have answered all the issues raised in the reviews, and that the paper is clearly better after incorporating the changes.

For your convenience, here is a list of the main changes:

    - Added omitted definitions: exponential family (Section 3.1), Gaussian mechanism, subsampling (Section 3.2).
    - Added some more clarifications and details (e.g. about data typically being a vector of reals in Section 3.1).
    - Added additional plot (Figure 5) on the effect of the number of local data partitions with local averaging to Appendix.
    - Added fixes (including more explicit refs to several theorems) to improve readability and make them more self-contained.
    - Moved Related work Section to beginning.
    - Fixed all notational overlaps, inconsistencies, and typos.

Best regards
authors

---

> ### Comment · Action_Editors · 2023-03-06
> **Apologies**
>
> Dear authors, apologies for the delay. I have just been substituted in as a replacement AE, as the previous one was unresponsive. I will try to get the process moving on this paper again.

---

### Decision · Action_Editors · 2023-03-18

**Recommendation:** Accept with minor revision

**Comment:**

The above evaluations describe the submission's relation to each of the acceptance criteria. Since both are passed, the paper should be accepted. One of the reviewers still had concerns related to the communications improvement, and so the authors should perform a minor revisions with this in mind.

**Audience:**

Sentiment regarding whether the paper was interesting was less positive, and there was significant discussion. I pointed to the acceptance criteria (https://jmlr.org/tmlr/acceptance-criteria.html) and asked the reviewers specifically to speak towards these points. None of the reviewers were positive. Some comments included that the paper was not very technically interesting, the contribution being limited, weak technical novelty, not providing actionable lessons, and does not provide generalizable insights to the community. Some of these comments were not relevant for this criteria (e.g., the directions specify to ignore things like "novelty") and thus not taken into account.

Despite all this, I believe that, by the criteria of TMLR, this work passes the bar. The interesting-ness of the paper may be low for the differential privacy (DP) community. However, for those interested in federated VI, and have concerns related to privacy in these settings, then this would be of interest. My hesitations are that a) the intersection of the TMLR community and the federated VI + privacy community (who are not already in the DP community) may be small (to the point of non-existence), and b) the level of interesting-ness might be low, if, as the reviewers claim, it is the straightforward and natural way to include DP in the training pipeline. That said, the acceptance criteria say: "Generally, a reviewer that is unsure as to whether a submission satisfies this criteria should assume that it does." Thus, I err on the side of satisfying this criteria.

**Claims And Evidence:**

The reviewers agreed that the paper is sound. This was not the case for the initial submission, and the reviewers had many questions and concerns. The authors admirably addressed almost all of them. One reviewer pointed out "the paper several times mentions communication improvement. Though authors said they added clarification (magnitude improvement), it is not explicit in theorems and experimental discussion," so this should be addressed.